# Conformal Prediction Sets with Improved Conditional Coverage using Trust Scores

## Abstract

Standard conformal prediction offers a marginal guarantee on coverage, but for prediction sets to be truly useful, they should ideally ensure coverage *conditional* on each test point. However, it is impossible to achieve exact, distribution-free conditional coverage in finite samples. In this work, we propose an alternative conformal prediction algorithm that targets coverage where it matters most—in instances where a classifier is *overconfident in its incorrect predictions*. We start by dissecting miscoverage events in marginally-valid conformal prediction, and show that miscoverage rates vary based on the classifier's confidence and its deviation from the Bayes optimal classifier. Motivated by this insight, we develop a variant of conformal prediction that targets coverage conditional on a reduced set of two variables: the classifier's confidence in a prediction and a nonparametric *trust score* that measures its deviation from the Bayes classifier. Empirical evaluation on multiple image datasets shows that our method generally improves conditional coverage properties compared to standard conformal prediction, including class-conditional coverage, coverage over arbitrary subgroups, and coverage over demographic groups.

## 1 Introduction

Machine learning models are envisioned to inform decision-making in high-stakes applications such as medical diagnosis (Kumar et al., 2023; Elfanagely et al., 2021; Caruana et al., 2015). Consequently, there is a critical need for actionable and useful uncertainty estimates to mitigate the risks associated with incorrect decisions influenced by overconfident models. Conformal prediction (Vovk et al., 2005) is a framework for constructing prediction sets that provide finite-sample *marginal coverage* guarantees without making any modeling or distributional assumptions beyond exchangeability; i.e., the sets contain the correct output with a probability specified by the user. Given a calibration dataset $\{(X_i, Y_i)\}_{i=1}^n$ and a new test point $(X_{n+1}, Y_{n+1})$, split conformal prediction (referred to as simply "conformal prediction" from here on) constructs a prediction set $\mathcal{C}(X_{n+1}) \subseteq \mathcal{Y}$ that satisfies,

$$\mathbb{P}(Y_{n+1} \in \mathcal{C}(X_{n+1})) \geq 1 - \alpha, \tag{1}$$

for $\alpha \in (0, 1)$. However, marginal validity does not necessarily ensure that the prediction sets are actionable in arbitrary contexts, as coverage can be unacceptably poor for individual predictions.

To highlight the limitations of marginal validity, consider a scenario where a model is used to diagnose a disease in a population where 90% of the cases are straightforward to diagnose, while 10% are challenging. Prediction sets that only cover in the straightforward cases may achieve 90% marginal coverage, but may be useless for the cases where uncertainty estimates are most critical for clinicians. Ideally, one would like to construct prediction sets that satisfy *conditional coverage*; i.e., $\mathbb{P}(Y_{n+1} \in \mathcal{C}(X_{n+1})|X_{n+1} = x) \geq 1 - \alpha, \forall x$. Unfortunately, however, it is well known that it is impossible to attain distribution-free exact conditional coverage in a meaningful sense (Lei & Wasserman, 2014; Vovk, 2012; Barber et al., 2021). The question becomes one of approximate conditional coverage.

In this work, we introduce a relaxed objective for conditional coverage and a variant of conformal prediction that ensures prediction sets adapt to the true uncertainty of test instances. Since the feature space $\mathcal{X} \subseteq \mathbb{R}^d$ can be high-dimensional, achieving approximate $X$-conditional coverage is challenging. Gibbs et al. (2023) proposed a method that can theoretically achieve conditional coverage with respect to any function class; however, implementing this with a function class that corresponds to $X$-conditional coverage is infeasible. Our key idea is to instead identify a lower-dimensional variable

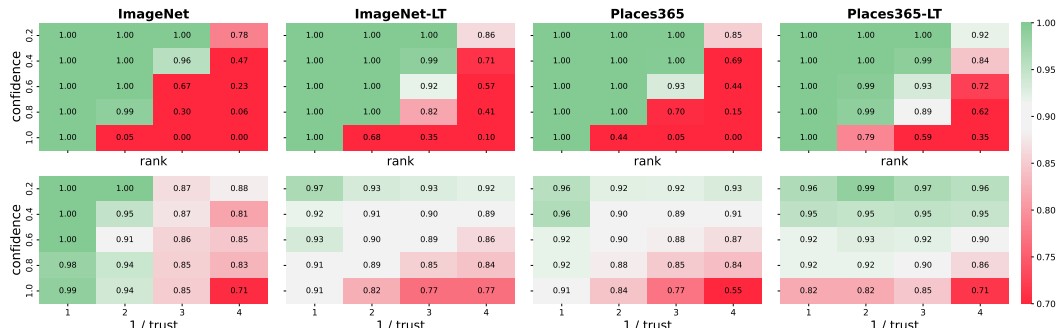

Figure 1: **Miscoverage patterns in standard conformal prediction.** Illustration of conditional coverage of standard conformal prediction (STANDARD) over regions of the feature space binned by model confidence and rank of the true class (top) (Conf × Rank), and confidence and *trust score* (bottom) (Conf × Trust). We set $\alpha = 0.1$; hence, red bins indicate undercoverage (coverage $< 0.9$) and green bins indicate overcoverage (coverage $> 0.9$). We split samples into equal-size bins based on rank and trust score. As a special case for ImageNet, we manually define the rank bins, as $\sim 75\%$ test samples have accurate predictions. Models are confidence calibrated using temperature scaling.

$V$, where the value of $V$ is indicative of whether standard conformal prediction will over- or under-cover the corresponding test instance. Based on $V$, we propose a function class that is both practical to implement and yields conditional guarantees important for decision-making. We then construct prediction sets that achieve (approximate) conditional coverage with respect to $V$, i.e.,

$$\mathbb{P}(Y_{n+1} \in \mathcal{C}(X_{n+1}) | V_{n+1} = v) \geq 1 - \alpha, \forall v. \tag{2}$$

Our choice of the variable $V$ is driven by an analysis of how miscoverage events are distributed across test instances in standard conformal prediction. In the classification setting, we select $V$ as a statistic that identifies instances where conformal prediction is most likely to fail—specifically when the classifier is *overconfident in its incorrect predictions*. Figure 1 illustrates the relationship between miscoverage rates in standard conformal prediction and the classifier's overconfidence as measured by its reported confidence (top softmax output) and the rank of $Y_{n+1}$ in its sorted softmax probabilities. The miscoverage patterns in Figure 1 suggest that these two factors are predictive of whether conformal prediction over- or under-covers a test instance. We defer a formal analysis to Section 2.2.

A variant of conformal prediction that achieves the relaxed conditional guarantee in (2) with respect to the statistic $V = \{\text{Conf}, \text{Rank}\}$—where Conf is the classifier's confidence and Rank is the rank of $Y_{n+1}$—ensures that the resulting prediction sets cover both low- and high-uncertainty test instances. Such a procedure would produce more balanced coverage patterns compared to standard conformal prediction. Since the Rank variable depends on $Y_{n+1}$, which is not available to us during test time, we propose a practical choice for $V$ that assesses the classifier's overconfidence through the model confidence (Conf) and a nonparametric *trust score* (Trust) that measures the disagreement of the model predictions with the Bayes-optimal classifier (Jiang et al., 2018) (Section 3.1).

We perform an extensive evaluation on four large-scale image classification datasets: ImageNet (Russakovsky et al., 2015), Places365 (Zhou et al., 2018), and their corresponding long-tail versions ImageNet-LT and Places365-LT (Liu et al., 2019). Since conditional coverage has not been studied in this setting previously, we propose a suite of evaluation metrics to measure approximate conditional coverage. We find that our proposed method reduces coverage gap across test instances as evaluated by these metrics in all datasets. We also perform evaluation on the Fitzpatrick 17k dataset (Groh et al., 2021) for skin condition classification in clinical images, where we are able to improve coverage across different skin types without access to type annotations.

### 1.1 RELATED WORK

**Group-conditional conformal prediction.** A widely adopted relaxation of conditional coverage in prior work is based on group-conditional guarantees of the form $\mathbb{P}(Y_{n+1} \in \mathcal{C}(X_{n+1}) \mid X_{n+1} \in G) \geq 1 - \alpha$ for all groups $G \in \mathcal{G}$ (Vovk et al., 2003; Barber et al., 2021; Jung et al., 2023). This concept is often motivated by the idea that instead of $X$-conditional coverage, one can ensure the coverage

guarantee holds over predetermined subgroups that might otherwise be underserved by marginal coverage (Romano et al., 2020a). Mondrian conformal prediction (Vovk et al., 2003) achieves exact group-conditional coverage in finite samples when the groups in $\mathcal{G}$ are disjoint. The procedure involves splitting the calibration data into subgroups and then separately calibrating on each group. Within this framework, Romano et al. (2020a) focus on achieving equal coverage over disjoint protected groups of interest. Barber et al. (2021) propose an approach that allows groups in $\mathcal{G}$ to overlap; however, this method can be highly conservative and result in wide prediction intervals that over-cover. To provide practical, "multivalid" coverage guarantees over arbitrary subgroups, (Jung et al., 2023) propose learning quantile multiaccurate predictors by minimizing the pinball loss over the class of functions $\mathcal{F} = \{\sum_{G \in \mathcal{G}} \beta_G \mathbb{1}\{x \in G\} : \beta \in \mathbb{R}^{|\mathcal{G}|}\}$. While Jung et al. (2023) provide PAC-style guarantees, Gibbs et al. (2023) propose a conditional conformal procedure that yields exact coverage guarantees over arbitrary collection of groups in finite samples, and also extends beyond the group setting to finite-dimensional classes. That said, as noted earlier, it is not feasible to run this method with a function class that can guarantee approximate $X$-conditional coverage. Our work is complementary to this line of work: we propose a practical way to construct a function class that guarantees an interpretable notion of conditional coverage broader than group-conditional coverage over pre-specified groups.

**Learning features from data for improved conditional coverage.** We note there are some recent works relevant to us that share similar motivation of learning features from data to improve conditional coverage; however our methodology presented herein differs greatly. Yuksekgonul et al. (2023) propose a density-based *atypicality* notion to improve calibration and conditional coverage with respect to input atypicality. They implement a special case of Mondrian conformal prediction to improve coverage in high atypicality or low confidence groups. In our work, the goal is to improve conditional coverage more generally beyond the chosen statistic $V$ and our evaluation reflects the same. Our method is also very different from Mondrian conformal prediction as we discuss. (Kiyani et al., 2024) propose to learn partitioning of the covariate space such that points in the same partition are similar in terms of their prediction sets in order to improve conditional validity. They present an algorithm that iteratively updates the partitioning and prediction sets over a given calibration data set. While they learn low-dimensional features from the calibration data, we study general patterns of miscoverage in standard conformal prediction and propose a two-dimensional statistic that consistently shows to be effective. It would be interesting to explore their approach in our evaluation setup that extends beyond pre-defined groups. In the regression setting, (Sesia & Romano, 2021) learn conditional histograms from the data to detect the skewness of $Y \mid X$ and estimate the quantiles of the conditional distribution. Guan (2022) propose a localized conformal prediction framework to adapt to the heterogeneity of the conditional distribution. They propose weighting the conformal scores differently based on the observed feature value $X_{n+1}$ of the test sample.

**Other approaches for improving conditional coverage.** With the goal of achieving approximate $X$-conditional coverage, previous works have proposed new conformity score functions (Romano et al., 2019; 2020b; Angelopoulos et al., 2021) that yield significant practical improvements. Ding et al. (2023) focus on achieving class-conditional coverage instead, and introduce a clustered conformal prediction method for $Y$-conditional coverage. As argued by Angelopoulos et al. (2021) and Ding et al. (2023), we note that in high-signal problems like image classification where $Y$ is perfectly determined by $X$, $X$-conditional coverage is less interpretable as an objective.

## 2 PRELIMINARIES

### 2.1 STANDARD CONFORMAL PREDICTION

In this paper, we consider a classification setting in which each input feature $X_i \in \mathcal{X}$ is associated with a class label $Y_i$ drawn from a discrete set of possible classes $\mathcal{Y}$. Let $s : \mathcal{X} \times \mathcal{Y} \to \mathbb{R}$ be a *conformity score* function that measures how well the label $y$ "conforms" to a model prediction at $x$, where lower scores indicate better agreement. (A simple choice for the score is $s(x, y) = 1 - \hat{f}_y(x)$, where $\hat{f} : \mathcal{X} \to \mathcal{Y}$ is a pretrained classifier and $\hat{f}_y(x)$ is its softmax output for class $y$). Given the calibration data set, $\{(X_i, Y_i)\}_{i=1}^n$, and a model $\hat{f}$, a conformal prediction set $\mathcal{C}(X_{n+1})$ for a test point $X_{n+1}$ is constructed by evaluating the conformity scores $s_i = s(X_i, Y_i), 1 \leq 1 \leq n$. Then, we compute $\hat{q}$ as

the $\lceil (n+1)(1-\alpha) \rceil /n$ empirical quantile of $\{s_i\}_{i=1}^n$, and use $\hat{q}$ to construct the prediction sets

$$\mathcal{C}(X_{n+1}) = \{y : s(X_{n+1}, y) \leq \hat{q}\}. \tag{3}$$

We refer to this as STANDARD split conformal prediction following Ding et al. (2023). STANDARD conformal prediction guarantees marginal validity as described in (1) as long as the calibration and test scores $s_1, \ldots, s_n, s_{n+1}$ are exchangeable. However, as discussed earlier, marginal coverage may be insufficient for $\mathcal{C}$ to be practically useful and we aim for a stronger notion of conditional coverage.

## 2.2 DISSECTING MISCOVERAGE PATTERNS IN STANDARD CONFORMAL PREDICTION

**Exact conditional coverage with an oracle classifier.** Imagine an "oracle" classifier $f^*$ which perfectly captures the distribution $f_y^*(x) = P(Y = y|X = x), \forall y \in \mathcal{Y}, x \in \mathcal{X}$. Romano et al. (2020b) showed that one could construct optimal prediction sets $\mathcal{C}^{\text{oracle}}(x)$ with exact conditional coverage by leveraging the *order statistics* $f_{(1)}^*(x) \geq f_{(2)}^*(x) \geq \cdots \geq f_{(|\mathcal{Y}|)}^*(x)$, for $\{f_y^*(x) : y \in \mathcal{Y}\}$, as follows:

$$\mathcal{C}^{\text{oracle}}(x) = \{\text{'}y\text{' indices of the } k \text{ largest } f_y^*(x)\}, \text{where } k = \inf\left\{ k' : \sum_{j=1}^{k'} f_{(j)}^*(x) \geq 1 - \alpha \right\}.$$

With knowledge of the true probabilities $\{f_y^*(x)\}_y$, $k = \inf\{k' : \sum_{j=1}^{k'} f_{(j)}^*(x) \geq 1 - \alpha\}$ can be thought of as a generalization of the conditional quantile function for continuous outcomes. In this sense, the oracle sets are conditionally valid because they correspond to the $(1-\alpha)$ quantile of $Y|X$. In practice, however, constructing such sets is impossible because we do not have access to $f^*$.

**Overconfidence and (conditional) miscoverage.** A common implementation of conformal prediction approximates the oracle algorithm above using $\hat{f}$ as a "plug-in" estimate of $f^*$ to compute the score

$$s(x, y) = \sum_{j=1}^k \hat{f}_{(j)}(x), \text{where } \hat{f}_{(k)}(x) = \hat{f}_y(x), \tag{4}$$

$\hat{f}_{(j)}(x)$ denotes the $j^{\text{th}}$ sorted value of the softmax outputs of $\hat{f}$, and $k$ is the index in the sorted order that corresponds to true class $y$ (Romano et al., 2020b; Angelopoulos et al., 2021). This variant of conformal prediction may empirically improve adaptivity of the resulting sets, but still guarantees only marginal validity in finite samples. This means some regions of the feature space will be over-covered, while others will be under-covered. Intuitively, miscoverage is more likely to cover instances where $\hat{f}$ is a poor approximation of the oracle $f^*$, which are also the cases where uncertainty quantification is most critical. To formalize this intuition, let the output of the classifier $\hat{f}$ be parameterized as

$$\hat{f}_j(z; T) = \frac{e^{z_j/T}}{\sum_{i=1}^{|\mathcal{Y}|} e^{z_i/T}}, \tag{5}$$

where $z \in \mathbb{R}^{|\mathcal{Y}|}$ are the model logits and $T \geq 1$ is a temperature parameter. Then, the following result provides insight into miscoverage pattern by STANDARD (proof is provided in Appendix A).

**Proposition 1.** *Let $\mathcal{C}(.)$ be constructed as in (3) for a classifier with the parameterization in (5), with the conformity score in (4). Let $\textsf{Rank}(X_{n+1}, Y_{n+1})$ be the position of softmax output $\hat{f}_{Y_{n+1}}(X_{n+1})$ in the decreasing sorted order of class probabilities $\hat{f}_{(1)}(X_{n+1}), \ldots, \hat{f}_{(|\mathcal{Y}|)}(X_{n+1})$, and $\textsf{Conf}(X_{n+1}) = \max_y \hat{f}_y(X_{n+1})$. Then, for any $\textsf{Rank}(X_{n+1}, Y_{n+1}) > 1$, if $\textsf{Conf}(x) > \textsf{Conf}'(x) (T < T')$, we have for a fixed $\hat{q}$*

$$\mathbb{P}(Y_{n+1} \in \mathcal{C}(X_{n+1}) \mid X_{n+1} = x, \textsf{Conf}(x)) \leq \mathbb{P}(Y_{n+1} \in \mathcal{C}(X_{n+1}) \mid X_{n+1} = x, \textsf{Conf}'(x)),$$

$\forall x \in \mathcal{X}$. The variables $\textsf{Rank}(x, y)$ and $\textsf{Conf}(x)$ measure a model's overconfidence—a model is considered overconfident when it assigns a high value on its largest softmax output while the true class ranks poorly in the descending order of predicted class probabilities. This result shows that, conditional on $X_{n+1} = x$, if the model's prediction is incorrect (i.e., $\textsf{Rank}(X_{n+1}, Y_{n+1}) > 1$), an increase in overconfidence leads to a reduction in coverage probability. This is because overconfident models concentrate the probability mass on fewer classes, yielding smaller prediction sets.

**Miscoverage patterns for STANDARD.** Proposition 1 highlights the impact of classifier confidence on *pointwise* coverage probabilities. In Figure 1, we empirically visualize the *marginal* coverage rates of STANDARD in different strata of $\textsf{Conf}$ and $\textsf{Rank}$ across various image classification datasets. As we can

see, the marginal version of the monotonic relationship in Proposition 1 holds consistently across all predictions that share the same value for Rank. Additionally, we observe that for a fixed Conf, coverage probability decreases as Rank increases. This finding suggests that, given the values of Conf and Rank for a test instance, we can determine its position within the strata in Figure 1, predict whether STANDARD will likely over- or under-cover the true label, and adjust the conformal prediction set accordingly to achieve more balanced coverage across test instances.

## 3 CONDITIONAL CONFORMAL PREDICTION WITH TRUST SCORES

Consider a scenario where the feature space $\mathcal{X}$ is discrete, and we have access to a large calibration set that includes all possible values in $\mathcal{X}$. In this case, one approach to constructing conditionally valid sets is to select a distinct threshold $\hat{q}$ in (3) for each $x \in \mathcal{X}$. This procedure would assign a larger $\hat{q}$ for instances where the model $\hat{f}$ is more prone to errors, and a smaller $\hat{q}$ where errors are less likely. However, in practice, this procedure is infeasible because $\mathcal{X}$ is typically high-dimensional or continuous. More generally, Lei & Wasserman (2014) and Barber et al. (2021) have shown that distribution-free conditionally valid predictive inference is impossible to attain meaningfully.

The key idea behind our proposed method is to condition on a lower-dimensional statistic $V \in \mathcal{V}$, rather than the full feature space $\mathcal{X}$, where $|\mathcal{V}| \ll |\mathcal{X}|$. We select this statistic as a proxy to identify test instances prone to error, enabling the selection of a distinct threshold $\hat{q}$ for each $v \in \mathcal{V}$, rather than conditioning on each individual point in $\mathcal{X}$. While this procedure does not achieve $X$-conditional coverage, it provides coverage conditioned on the likelihood of under-coverage by STANDARD. The miscoverage patterns discussed in Section 2.1 motivate the selection of $V$ as $V = \{\text{Conf}, \text{Rank}\}$. Consequently, instead of strict conditional coverage, $\mathbb{P}(Y_{n+1} \in \mathcal{C}(X_{n+1}) \mid X_{n+1} = x) \geq 1 - \alpha$, for all $x \in \mathcal{X}$, we adopt a more relaxed notion of conditional coverage as follows:

$$\mathbb{P}(Y_{n+1} \in \mathcal{C}(X_{n+1}) \mid \text{Conf}(X_{n+1}) = c, \text{Rank}(X_{n+1}, Y_{n+1}) = r) \geq 1 - \alpha, \qquad (6)$$

for all $c \in [0, 1]$ and $r \in \{1, \ldots, |\mathcal{Y}|\}$. However, the label $Y_{n+1}$ is not available at test time, and thus $\text{Rank}(X_{n+1}, Y_{n+1})$ cannot be used to construct the prediction sets. In the next section, we propose an alternative to the $\text{Rank}(X, Y)$ variable that can be computed using calibration data.

### 3.1 IMPLEMENTATION USING TRUST SCORES

The $\text{Rank}(X, Y)$ variable measures how far from the top softmax score a classifier ranks the true class $Y$ for a given input $X$. Since we do not have access to the label $Y$, we use the *trust score* proposed in Jiang et al. (2018) as a proxy for Rank. The trust score is a nonparametric statistic that measures the agreement between the classifier $\hat{f}$ and the Bayes-optimal classifier on a given testing point $X$. Formally, the trust score $\text{Trust}(X; \hat{f})$ for a classifier $\hat{f}$ on test point $X$ is defined as

$$\text{Trust}(x; \hat{f}) := d\left(x, \widehat{H}_\delta(P_{\tilde{y}})\right) / d\left(x, \widehat{H}_\delta(P_{\hat{y}})\right), \qquad (7)$$

where $\widehat{H}_\delta(P) := \{x \in X : r_k(x) \leq \varepsilon\}$; $k$-NN radius $r_k(x) := \inf\{r > 0 : |B(x, r) \cap X| \geq k\}$, $\varepsilon := \inf\{r > 0 : |\{x \in X : r_k(x) > r\}| \leq \delta \cdot n\}$. The evaluation of Trust proceeds in two stages: first, a $\delta$-*high-density-set* $\widehat{H}_\delta(P_\ell)$ (for continuous density function $P$ with compact support $\mathcal{X}$) is estimated for each class $\ell$ from the training data by filtering out $\delta$-fraction of samples with lowest density. Then, for a given test sample, the trust score ($\text{Trust}(X)$) (7) is computed as the ratio of the Euclidean distance between $X$ and the nearest point in the training set with class label *different* from the top-1 predicted label (say, $\tilde{y}$), and the distance between $X$ and the nearest point with class label *same* as the top-1 predicted label by the classifier (say, $\hat{y}$). We provide the implementation details in Appendix B.3.

Theorem 4 in Jiang et al. (2018) provides the following guarantee for the trust score: for labeled data distributions with well-behaved class margins, when the trust score is large, the classifier likely agrees with the Bayes optimal classifier $\arg\max_{\ell \in \mathcal{Y}} \mathbb{P}(y = \ell | x)$, and when the trust score is small, the classifier likely disagrees with it. Given the Bayes-optimal classifier has low error, high probability of agreement with the Bayes-optimal classifier can help identify correct predictions ("trustworthy" examples), whereas high probability of disagreement can indicate the classifier is making an unreasonable decision. In our previous result, we showed that when $\text{Rank}(X, Y) > 1$, coverage decreases

with higher confidence. In this regard, the trust score can be used to signal when the true class has a higher probability of not being the top class in the model output probabilities. We present an empirical analysis of the correlation between Trust and Rank in Appendix C.3.

## 3.2 ALGORITHM

We introduce our CONDITIONAL method for conformal prediction with the goal of achieving approximate conditional coverage over the reduced variable set $V = \{\mathsf{Conf}(X_{n+1}), \mathsf{Trust}(X_{n+1})\}$:

$$\mathbb{P}(Y_{n+1} \in \mathcal{C}(X_{n+1}) \mid \mathsf{Conf}(X_{n+1}) \in I_1, \mathsf{Trust}(X_{n+1}) \in I_2) \geq 1 - \alpha, \tag{8}$$

where sub-intervals $I_1$ and $I_2$ are some discretization of $[0, 1]$ and $(0, \infty)$ respectively. The exact conditional coverage guarantee $\mathbb{P}(Y_{n+1} \in \mathcal{C}(X_{n+1}) | X_{n+1} = x) = 1 - \alpha, \ \forall x \in \mathcal{X}$, is equivalent to a marginal guarantee over all measurable functions $f$:

$$\mathbb{E}[f(X_{n+1}) \cdot (\mathbb{1}\{Y_{n+1} \in \mathcal{C}(X_{n+1})\} - (1 - \alpha))] = 0, \quad \text{for all measurable } f. \tag{9}$$

If $f(x) = x \mapsto 1$, we recover marginal coverage. Gibbs et al. (2023) propose a relaxation of the exact conditional coverage guarantee over all measurable $f$ to all $f$ belonging to some function class $\mathcal{F}$,

$$\mathbb{E}[f(X_{n+1}) \cdot (\mathbb{1}\{Y_{n+1} \in \mathcal{C}(X_{n+1})\} - (1 - \alpha))] = 0, \quad \text{for all } f \in \mathcal{F}. \tag{10}$$

A special case of this guarantee is group-conditional coverage; i.e., $\mathbb{P}(Y_{n+1} \in \mathcal{C}(X_{n+1}) \mid X_{n+1} \in G) = 1 - \alpha$ for all $G$ belonging to some collection of groups $\mathcal{G}$, where $\mathcal{F} = \{\sum_{G \in \mathcal{G}} \beta_G \mathbb{1}\{x \in G\} : \beta \in \mathbb{R}^{|\mathcal{G}|}\}$. The above conditional coverage guarantee can be achieved by fitting an augmented quantile regression problem over $\mathcal{F}$ where the unknown conformity score $s_{n+1}$ is imputed as $s$. The quantile estimate $\hat{g}_s$ is fit using the *pinball loss* $\ell_\alpha(g(X_i), s_i)$ as follows

$$\hat{g}_s = \underset{g \in \mathcal{F}}{\arg\min} \ \frac{1}{n+1} \sum_{i=1}^{n} \ell_\alpha(g(X_i), s_i) + \frac{1}{n+1} \ell_\alpha(g(X_{n+1}), s), \tag{11}$$

where $\ell_\alpha(g(X_i), s_i) = (1 - \alpha)(s_i - g(X_i))_+ + \alpha(g(X_i) - s_i)_+$. We compute the prediction set by

$$\mathcal{C}(X_{n+1}) = \{y : s(X_{n+1}, y) \leq \hat{g}_{s(X_{n+1}, y)}(X_{n+1})\}. \tag{12}$$

For a finite-dimensional linear class $\mathcal{F} = \{\Phi(\cdot)^\top \beta : \beta \in \mathbb{R}^d\}$ over the basis $\Phi : \mathcal{X} \to \mathbb{R}^d$, Gibbs et al. (2023) show that we can achieve an upper bound on coverage in (10) and the exact coverage guarantee with appropriate randomization.

**Theorem 1** (Theorem 2 Gibbs et al. (2023)). *Suppose* $\{(X_i, Y_i)\}_{i=1}^{n+1}$ *are independent and identically distributed. Let* $\mathcal{F} = \{\Phi(\cdot)^\top \beta : \beta \in \mathbb{R}^d\}$ *denote the class of linear functions over the basis* $\Phi : \mathcal{X} \to \mathbb{R}^d$. *If the distribution of* $s \mid X$ *is continuous, then for all* $f \in \mathcal{F}$, *we have*

$$|\mathbb{E}[f(X_{n+1}) \cdot (\mathbb{1}\{Y_{n+1} \in \mathcal{C}(X_{n+1})\} - (1 - \alpha))]| \leq \frac{d}{n+1} \mathbb{E}\left[\max_{1 \leq i \leq n+1} |f(X_i)|\right].$$

To achieve our coverage objective (8), we now define a function class $\mathcal{F}$ that depends on $V$.

**Corollary 1.** *Let* $\mathcal{F} = \{\sum_{I_1 \in \mathcal{I}_1} \beta_{1_{I_1}} \mathbb{1}\{\mathsf{Conf}(x) \in I_1\} + \sum_{I_2 \in \mathcal{I}_2} \beta_{2_{I_2}} \mathbb{1}\{\mathsf{Trust}(x) \in I_2\} : \beta_1 \in \mathbb{R}^{|\mathcal{I}_1|}, \beta_2 \in \mathbb{R}^{|\mathcal{I}_2|}\}$ *for some finite collection of sub-intervals* $\mathcal{I}_1, \mathcal{I}_2$. *Then,*

$$\mathbb{P}(Y_{n+1} \in \mathcal{C}(X_{n+1}) \mid \mathsf{Conf}(X_{n+1}) \in I_1, \mathsf{Trust}(X_{n+1}) \in I_2) \geq 1 - \alpha.$$

*If we randomize the non-conformity scores* $s$, *we have an upper bound on coverage,*

$$\mathbb{P}(Y_{n+1} \in \mathcal{C}(X_{n+1}) \mid \mathsf{Conf}(X_{n+1}) \in I_1, \mathsf{Trust}(X_{n+1}) \in I_2) \leq 1 - \alpha +$$

$$\frac{|\mathcal{I}_1| + |\mathcal{I}_2|}{(n+1)\mathbb{P}(\mathsf{Conf}(X_{n+1}) \in I_1, \ \mathsf{Trust}(X_{n+1}) \in I_2)}. \tag{13}$$

Note that with $\mathcal{F}$ as defined above in Corollary 1, we achieve the conditional coverage guarantee over $V$ in (8). However, this will be highly computationally inefficient when computing the prediction set (12) as $\mathcal{I}_1, \mathcal{I}_2$ can be very large. In order to speed up computation we use polynomial functions of $\mathsf{Conf}(X)$ and $\mathsf{Trust}(X)$, with the intent of capturing higher-order interactions between the conformity score function and $V$. To this end, we define $\mathcal{F}$ as a function class of degree-$d$ polynomials,

$$\Phi(X) = \{\mathsf{Conf}(X)^i \cdot \mathsf{Trust}(X)^j \mid i + j \leq d, i, j \geq 0\} \ ; \ \mathcal{F} = \{\Phi(\cdot)^\top \beta : \beta \in \mathbb{R}^{\frac{(d+1)(d+2)}{2}}\}. \tag{14}$$

If we are willing to allow $\mathcal{C}(X_{n+1})$ to be randomized, we can achieve exact coverage equal to $1 - \alpha$ (stated formally in Theorem 2, Appendix A). However, it is not desirable to have non-deterministic prediction sets in most practical scenarios. Hence, we use a non-randomized version in our experiments, which guarantees that prediction sets have at least $1 - \alpha$ coverage over $\mathcal{F}$.

## 4 EXPERIMENTS

We evaluate CONDITIONAL empirically on four large-scale image classification datasets and a clinical dataset in dermatology. We propose evaluation metrics that measure the gap in coverage from the desired $1 - \alpha$ level across different regions of the distribution. We also introduce a NAIVE baseline that aims to improve coverage with respect to $V$ via group-conditional coverage guarantees over non-overlapping subgroups in $V$ (Section 4.2). We evaluate STANDARD, NAIVE, and CONDITIONAL on all datasets, and show that CONDITIONAL achieves the best conditional coverage performance across all settings. CONDITIONAL also improves class-conditional coverage on all but one dataset.

### 4.1 EVALUATION METRICS

To evaluate approximate conditional coverage over test points $\{(X_{i'}, Y_{i'})\}_{i'=1}^N$, we propose binning the feature space into $|B|$ bins and computing the average coverage gap (CovGap) across these bins (15). The coverage gap measures the $l_1$ distance between the achieved coverage and the desired $1 - \alpha$ level across all bins. Here, $\hat{c}_b$ denotes the mean empirical coverage in bin $b$; i.e., $\hat{c}_b = (\sum_{X_{i'} \in b} \mathbb{1}\{Y_{i'} \in \mathcal{C}(X_{i'})\})/|b|$, where $|b|$ is the number of samples in bin $b$. This metric is inspired from the class coverage gap in Ding et al. (2023):

$$\text{CovGap} = \frac{1}{|B|} \sum_{b \in B} |\hat{c}_b - (1 - \alpha)| \times 100. \tag{15}$$

For comprehensive evaluation, we measure the conditional coverage performance using CovGap over three different binning schemes:

1. Conf $\times$ Trust: We apply two-dimensional binning by splitting the samples into evenly spaced bins based on Conf and then splitting each confidence bin into equal-size bins based on Trust score (see Figure 1 (bottom) for reference).

2. Conf $\times$ Rank : We apply two-dimensional binning by splitting the samples into evenly spaced bins based on Conf and then splitting each confidence bin into equal-size bins based on Rank (see Figure 1 (top) for reference).

3. Class-conditional: We split samples into bins based on their class labels, hence $|B| = |\mathcal{Y}|$.

We only consider bins with a non-zero number of samples ($|b| > 0$) in our evaluation. We also report marginal coverage and average set size ($\sum_{i'=1}^N |\mathcal{C}(X_{i'})|/N$) in our results.

### 4.2 EXPERIMENTAL SETUP

**Datasets.** We perform experiments on ImageNet (Russakovsky et al., 2015), Places365 (Zhou et al., 2018), and their corresponding long-tail versions ImageNet-LT and Places365-LT (Liu et al., 2019). ImageNet-LT and Places365-LT are constructed from the original datasets using a Pareto distribution with a power value $\alpha = 6$. Places365-LT has higher class imbalance than ImageNet-LT, defined by the number of samples in the largest class divided by the number of samples in the smallest class. We also evaluate on the Fitzpatrick 17k dataset (Groh et al., 2021) for skin disease diagnosis to study coverage of prediction sets across skin types (Section 4.4).

**Baselines.** Along with STANDARD split conformal prediction, we additionally include a NAIVE baseline that also aims to provide coverage with respect to $V$ using the Mondrian conformal prediction procedure (Vovk et al., 2003) discussed earlier. NAIVE fits a separate quantile $\hat{q}_b$ for each individual bin $b$ in the Conf $\times$ Trust binning setting, and the prediction sets for $X_{i'} \in b$ are computed as

$$\mathcal{C}(X_{i'}) = \{y : s(X_{i'}, y) \leq \hat{q}_b\}.$$

The objective is to evaluate the effectiveness of covering over a higher-dimensional function class $\mathcal{F}$.

**Experimental details.** We use a non-randomized version of the APS score (Romano et al., 2020b) (described in Section 2.1), as our conformity score: $s(x, y) = \sum_{j=1}^{k-1} \hat{f}_{(j)}(x)$, where $\hat{f}_{(k)}(x) = \hat{f}_y(x)$. We exclude $\hat{f}_y(x)$ to achieve smaller set sizes overall (Gibbs et al., 2023). We perform an extra step of temperature scaling to rescale the probabilities following past work (Angelopoulos et al., 2021; Guo et al., 2017). We consider $\alpha = 0.1$ for all experiments to achieve a desired coverage level of 90%. Further experimental details are provided in Appendix B.

Table 1: Conditional coverage evaluation on ImageNet, ImageNet-LT, Places365, and Places365-LT. Bold indicates the best (within $\pm 0.1$) coverage gap. We report standard errors in parentheses.

| | | Marginal | | Conditional | | |
| | | Coverage | Size | Conf × Trust CovGap | Conf × Rank CovGap | Class-conditional CovGap |
| Dataset | Method | | | | | |
| --- | --- | --- | --- | --- | --- | --- |
| ImageNet | STANDARD | 0.90 (0.00) | 4.32 (0.04) | 6.35 (0.11) | 33.12 (0.15) | 7.23 (0.04) |
| | NAIVE | 0.93 (0.00) | 7.21 (0.05) | 8.01 (0.50) | 23.82 (0.15) | 6.68 (0.02) |
| | CONDITIONAL | 0.90 (0.00) | 22.36 (0.83) | **4.37** (0.09) | **23.66** (0.24) | **6.02** (0.04) |
| ImageNet-LT | STANDARD | 0.89 (0.00) | 50.35 (0.02) | 4.47 (0.02) | 19.92 (0.03) | 8.32 (0.00) |
| | NAIVE | 0.89 (0.00) | 46.63 (0.07) | 2.99 (0.01) | 18.27 (0.02) | 8.29 (0.01) |
| | CONDITIONAL | 0.90 (0.00) | 58.64 (0.18) | **2.21** (0.01) | **17.09** (0.02) | **7.92** (0.00) |
| Places365 | STANDARD | 0.90 (0.00) | 14.17 (0.07) | 5.78 (0.08) | 25.58 (0.09) | **4.99** (0.05) |
| | NAIVE | 0.90 (0.00) | 12.76 (0.11) | **2.40** (0.16) | 22.22 (0.13) | 5.11 (0.06) |
| | CONDITIONAL | 0.90 (0.00) | 15.98 (0.57) | 4.38 (0.10) | **22.09** (0.12) | **4.98** (0.05) |
| Places365-LT | STANDARD | 0.90 (0.00) | 43.46 (0.10) | 5.55 (0.02) | 13.23 (0.01) | **5.34** (0.00) |
| | NAIVE | 0.90 (0.00) | 38.54 (0.06) | 4.55 (0.02) | **11.75** (0.01) | 5.61 (0.01) |
| | CONDITIONAL | 0.90 (0.00) | 37.07 (0.03) | **1.72** (0.00) | **11.75** (0.01) | 5.65 (0.00) |

## 4.3 RESULTS

Table 1 presents the CovGap of STANDARD, NAIVE, and CONDITIONAL on all datasets. We see that while all methods achieve the desired marginal coverage level of 90%, there is a significant difference in conditional coverage as evaluated by the coverage gap. Overall, CONDITIONAL achieves the best performance (lowest CovGap) across different binning schemes in nearly all experimental settings.

Focusing our attention on the Conf × Trust and Conf × Rank CovGap metrics, we observe that CONDITIONAL significantly outperforms STANDARD on all datasets, demonstrating improved conditional coverage. CONDITIONAL also performs better or comparable to NAIVE on the oracle metric Conf × Rank CovGap, which shows the effectiveness of our polynomial function class $\mathcal{F}$ (14). Note that CONDITIONAL has lower Conf × Trust coverage gap than NAIVE on all but one dataset, despite NAIVE having access to Conf × Trust bin information during the calibration phase. The improved performance of CONDITIONAL compared to STANDARD as well as NAIVE empirically validates our choice of $V$. We can see this improved conditional coverage is achieved with sets that are not significantly larger than STANDARD (and, in fact, smaller in Places365-LT).

Further, we also evaluate class-conditional coverage gap of all methods. Despite not explicitly targeting class-conditional coverage, our method generally improves the Class-conditional CovGap over STANDARD and NAIVE.

**Approximate $X$-conditional coverage.** Beyond evaluating coverage over the well-defined axes of Conf × Trust and Conf × Rank , we also measure approximate conditional coverage via a notion of local coverage over $\ell_2$ balls in the feature space (Barber et al., 2021) (Figure 2). We evaluate the coverage gap between randomly sampled Euclidean balls of a fixed radius $r$, varying $r$ as shown in the figure from $[r_{min}, r_{max}]$ (see Appendix B.4 for further details on the choice of $r$ and simulation parameters). The CovGap in Figure 2 shows that CONDITIONAL generally performs better than STANDARD and NAIVE when $r$ is relatively small. This is an approximation of local coverage as the neighborhood is small. As $r$ approaches $r_{max}$, the coverage for individual balls approach marginal coverage, and hence the CovGap decreases and the performance of all methods is typically comaparable. These results indicate that our coverage objective and method make significant progress towards improving conditional coverage in classification settings.

## 4.4 FITZPATRICK 17K DATASET: SKIN CONDITION CLASSIFICATION IN CLINICAL IMAGES

Fitzpatrick 17k (Groh et al., 2021) is a dataset of clinical images with skin condition labels and skin type labels 1 through 6 based on the Fitzpatrick scoring system. The Fitzpatrick skin type labels are annotated by a team of humans, and a small subset of images with annotated disagreement are

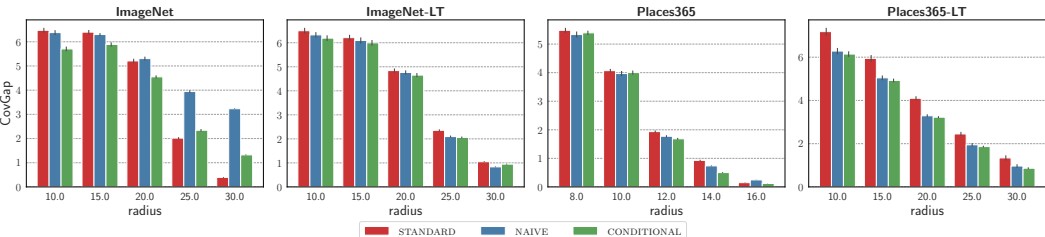

Figure 2: **Average coverage gap between randomly sampled Euclidean balls of fixed radius.** We vary the radius on x-axis. Standard errors are reported by error bars.

Table 2: Conditional coverage evaluation on Fitzpatrick 17k. Bold indicates the best (within $\pm 0.1$) coverage gap and worst-group coverage. We report standard errors in parentheses.

| | Marginal | | Conditional | | |
|---|---|---|---|---|---|
| | Coverage | Size | Skin type-conditional CovGap | Worst-group coverage | Class-conditional CovGap |
| **Method** | | | | | |
| STANDARD | 0.90 (0.00) | 27.30 (0.12) | 1.88 (0.13) | 0.86 (0.01) | 7.69 (0.10) |
| GROUPWISE | 0.90 (0.00) | 27.53 (0.21) | **1.76** (0.18) | 0.86 (0.01) | **7.60** (0.18) |
| CONDITIONAL | 0.90 (0.00) | 30.17 (0.14) | **1.73** (0.14) | **0.87** (0.01) | **7.47** (0.13) |

labeled as unknown. Higher Fitzpatrick skin type label indicates darker skin tones. The dataset has significantly fewer images of dark skin types compared to light skin, accompanied by imbalance of skin condition labels across skin types. Past work has shown there is disparity in model performance across skin tones and worse performance on uncommon diseases (Daneshjou et al., 2022; Yuksekgonul et al., 2023). Fitzpatrick 17k has 114 skin conditions (classes). Further details on the dataset and the experiment are provided in Appendix B.2.

We evaluate how our proposed method reduces the coverage gap across skin type groups *without access to type labels* (Table 2). For this specific setup, we naturally consider skin type groups as our evaluation bins. Similar to NAIVE, we include an analogous GROUPWISE baseline that has access to skin type annotations and computes an individual quantile level $\hat{q}$ for each group. We report the Skin type-conditional CovGap to measure the coverage gap between groups, along with Worst-group coverage. We see that CONDITIONAL reduces the coverage gap as well as improves worst-group coverage over STANDARD. Despite having no access to group labels, CONDITIONAL performs better or comparable to GROUPWISE, while having lesser variance. We also achieve lower class-conditional CovGap than STANDARD with GROUPWISE and CONDITIONAL.

## 5 DISCUSSION

Conformal prediction guarantees exact coverage *marginally* across test samples in finite samples. However, for uncertainty sets to be truly meaningful and useful, they should provide coverage *conditional* on test instances where uncertainty is higher or where the model is more likely to err. Unfortunately, achieving $X$-conditional coverage in finite samples is impossible without making distributional assumptions. In this paper, we propose a relaxed notion of conditional coverage that improves the practical utility of prediction sets by ensuring coverage where it matters most—specifically, in cases where a classifier is overconfident in its incorrect predictions. To assess a classifier's overconfidence, we use its reported confidence (softmax probabilities) in combination with a nonparametric *trust score* that measures its alignment with the Bayes classifier. We develop a practical variant of conformal prediction that achieves approximate conditional coverage with respect to these two variables, and demonstrate that it improves conditional coverage properties in a general sense, including subgroup-level and class-conditional coverage. By reducing the coverage gap across relevant subpopulations, our resulting prediction sets can lead to fairer and improved downstream decision-making, especially in high-stakes applications where miscoverage can be consequential.

While we show trust scores significantly improve conditional coverage in conformal prediction, they also come with limitations. Particularly, in cases where the trust scores are not a good approximate of the rank of the true class, our method may not improve conditional coverage properties over standard conformal prediction. To add, our function class is susceptible to computational difficulties at higher polynomial degrees beyond a threshold. Future work can explore more sophisticated function classes to achieve our proposed conditional coverage objective.

## 6 REPRODUCIBILITY STATEMENT

We provide the code to reproduce our experiments as part of the supplemental material. We also describe all experimental details in Appendix B including dataset and model details, experimental setup, and details on evaluation metrics. We include proofs for our theoretical results in Appendix A.

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

## A PROOFS

*Proof of Proposition 1.* We parameterize the output of the classifier $\hat{f}$ as

$$\hat{f}_j(z; T) = \frac{e^{z_j/T}}{\sum_{i=1}^{|\mathcal{Y}|} e^{z_i/T}},$$

where $z \in \mathbb{R}^{|\mathcal{Y}|}$ are the model logits and $T \geq 1$ is a temperature parameter.

Then, for any $y' \in [K]$ and any $T$, let $\mathcal{C}(z; T)$ be defined as in equation 3 with the conformity score defined as in (4) parameterized as

$$s(z, y'; T) = \sum_{j=1}^{k} \hat{f}_{(j)}(z; T), \text{where } \hat{f}_{(k)}(z; T) = \hat{f}_{y'}(z; T), \tag{16}$$

mutatis mutandis. Then, for any $T' > T$,

$$\mathcal{C}(z; T) \subseteq \mathcal{C}(z; T'). \tag{17}$$

In particular, this means that for any $y \in [K]$,

$$\mathbb{1}(y \in \mathcal{C}(z; T)) \implies \mathbb{1}(y \in \mathcal{C}(z; T')), . \tag{18}$$

and thus that for any joint distribution $P$ over $(z, y)$, that

$$\mathbb{P}_P(y \in \mathcal{C}(z; T) \mid z) \leq \mathbb{P}_P(y \in \mathcal{C}(z; T') \mid z) \tag{19}$$

and

$$\mathbb{P}_P(y \in \mathcal{C}(z; T)) \leq \mathbb{P}_P(y \in \mathcal{C}(z; T')). \tag{20}$$

*Proof of Theorem 1.* See Theorem 2, (Gibbs et al., 2023).

*Proof of Corollary 1.* Corollary 1 follows directly from Theorem 1 in the special case where we choose $\mathcal{F} = \{\sum_{I_1 \in \mathcal{I}_1} \beta_{1_{I_1}} \mathbb{1}\{\mathsf{Conf}(x) \in I_1\} + \sum_{I_2 \in \mathcal{I}_2} \beta_{2_{I_2}} \mathbb{1}\{\mathsf{Trust}(x) \in I_2\} : \beta_1 \in \mathbb{R}^{|\mathcal{I}_1|}, \beta_2 \in \mathbb{R}^{|\mathcal{I}_2|}\}$.

**Theorem 2** (Proposition 4 Gibbs et al. (2023)). *Suppose $\{(X_i, Y_i)\}_{i=1}^{n+1}$ are independent and identically distributed. Let $\mathcal{F} = \{\Phi(\cdot)^\top \beta : \beta \in \mathbb{R}^d\}$ denote the class of linear functions over the basis $\Phi : \mathcal{X} \to \mathbb{R}^d$. If we optimize the dual formulation of (11) and the dual solutions are computed using an algorithm that is symmetric in the input data, then the randomized prediction set $\mathcal{C}_{\mathrm{rand}}(X_{n+1})$ achieves exact coverage for all $f \in \mathcal{F}$:*

$$\mathbb{E}[f(X_{n+1}) \cdot (\mathbb{1}\{Y_{n+1} \in \mathcal{C}_{\mathrm{rand}}(X_{n+1})\} - (1 - \alpha))] = 0.$$

*Proof of Theorem 2.* See Proposition 4, (Gibbs et al., 2023). We formally state this result here to show that appropriate randomization of $\mathcal{C}(X_{n+1})$ can guarantee exact coverage without the continuity assumption on $s|X$ as in Theorem 1.

## B EXPERIMENTAL DETAILS

### B.1 EXPERIMENTAL SETUP

We set $\alpha = 0.1$ for our empirical evaluation. We run all our experiments with 10 random seeds $\{1, \ldots, 10\}$ and report the standard errors in our results. The randomness in our experiments is over splitting the validation set into calibration and evaluation data and fitting the temperature parameter (Guo et al., 2017).

**Evaluation Metrics.** For our conditional coverage evaluation, we use the notion of CovGap (inspired by class coverage gap in Ding et al. (2023)) to measure the coverage gap across multiple bins in the feature space. We propose three binning schemes: Conf × Trust, Conf × Rank, and Class-conditional CovGap. To apply two-dimensional binning, we split the samples into evenly spaced bins based on Conf and then split each confidence bin into equal-size bins based on Trust score or Rank depending on the binning scheme. For both two-dimensional binning schemes, we split the samples into 10 evenly spaced confidence bins and then 4 equal-size bins based on trust score or rank. We choose this splitting to have appreciable granularity while also ensuring most bins have sufficient number of samples in all cases. As a special case for ImageNet, we manually edit the Rank bins as $\sim 75\%$ test samples have accurate predictions.

## B.2 Datasets and Models

We follow the data processing steps and pretrained models used by Yuksekgonul et al. (2023) in our evaluation for ImageNet, ImageNet-LT, and Places365-LT.

**ImageNet.** ImageNet (Russakovsky et al., 2015) is a large-scale image classification dataset with 1000 classes. ImageNet has roughly balanced class distributions. We use the ImageNet-1k version from Torchvision (Marcel & Rodriguez, 2010) and the pretrained ResNet50 model. We split the validation dataset evenly into calibration and evaluation splits based on the random seed.

**ImageNet-LT.** ImageNet-LT is the long-tailed version of ImageNet with 1000 classes (Liu et al., 2019). ImageNet-LT was constructed from the original dataset using a Pareto distribution with a power value $\alpha = 6$, and has a maximum of 1280 images per class and minimum of 5 images per class. We use the validation split as our calibration set and use the test set for evaluation. Following Yuksekgonul et al. (2023), we use the ResNeXt50 model trained on ImageNet-LT by (Zhisheng Zhong & Jia, 2021) in our experiments.

**Places365.** Places365 (Zhou et al., 2018) contains 10 million images from 365 classes. For Places365, we train a ResNet152 model on our own (pretrained on ImageNet). We fine-tune on the train split of the original dataset and split the validation set evenly into calibration and test splits.

**Places365-LT.** Places365-LT is the long-tailed version of Places365 (Liu et al., 2019). Places365-LT was constructed from the original dataset using a Pareto distribution with a power value $\alpha = 6$, and has a maximum of 4980 images per class and a minimum of 5 images per class. We use the validation split as our calibration set and use the test set for evaluation. Following Yuksekgonul et al. (2023), we use the ResNet152 model trained on Places365-LT by (Zhisheng Zhong & Jia, 2021) in our experiments.

**Fitzpatrick 17k.** Fitzpatrick 17k (Groh et al., 2021) is a dataset of clinical images with skin condition labels and skin type labels 1 through 6 based on the Fitzpatrick scoring system. We use the training script provided by Groh et al. (2021) to train a ResNet18 model (pretrained on ImageNet) for 50 epochs. We consider the full classification task over 114 skin condition labels. For our group-conditional coverage evaluation, we consider Fitzpatrick skin types 1 through 6 as well as the Unknown type as our subgroups.

## B.3 Details on $\mathcal{F}$

We define our function class as $\mathcal{F} = \{\sum_{I_1 \in \mathcal{I}_1} \beta_{1_{I_1}} \mathbb{1}\{\mathsf{Conf}(x) \in I_1\} + \sum_{I_2 \in \mathcal{I}_2} \beta_{2_{I_2}} \mathbb{1}\{\mathsf{Trust}(x) \in I_2\} : \beta_1 \in \mathbb{R}^{|\mathcal{I}_1|}, \beta_2 \in \mathbb{R}^{|\mathcal{I}_2|}\}$. To compute $\mathsf{Trust}(x)$, we use features from the model's penultimate layer. Following Jiang et al. (2018), we skip the initial filtering step of the trust score algorithm to increase computational efficiency. For calculating the nearest neighbor distance to each class for the trust scores computation, we use IndexFlatL2 from FAISS (Johnson et al., 2017), Meta's open-sourced GPU-accelerated library for efficient similarity search. This reduces the single nearest neighbor search time to $\sim 0.06$ ms/sample (averaged over 10 runs) on ImageNet. For all experiments, we run CONDITIONAL with polynomial degree $d = 5$. We study the effect of $d$ in Appendix C.2. We would like to mention that with $d > 5$, the convex optimization procedure for computing the

prediction sets significantly slows down, and hence we do not report results for polynomial degree values $d > 5$.

### B.4 DETAILS ON APPROXIMATE X-CONDITIONAL COVERAGE EVALUATION

We share details regarding the experimental setup for evaluating approximate $X$-conditional coverage (Figure 2). In this experiment, we evaluate the coverage gap between randomly sampled $\ell_2$ balls in the feature space. We compute the Euclidean distance between features $X_i$ and $X_j$ from the model's penultimate layer. We vary the radius $r$ of the $\ell_2$ balls, where $r$ is evenly spaced in the interval $[r_{min}, r_{max}]$. $r_{min}$ and $r_{max}$ are estimated as the minimum and 90th percentile of the distribution of distances between randomly sampled pairs of points for each dataset, respectively. For a fixed $r$, we randomly sample 100 test points and find Euclidean balls with radius $r$ around the test point such that every ball should have at least 10 neighboring points. Then, we compute the coverage gap over these regions using Eq. 15. We perform 100 trials of this procedure for each $r$ for all datasets, and report the standard errors by error bars.

## C ADDITIONAL EXPERIMENTAL RESULTS

### C.1 MISCOVERAGE PATTERNS IN STANDARD AND CONDITIONAL

To demonstrate the improvement in conditional coverage over standard conformal prediction, we show the miscoverage patterns as in Figure 1 for CONDITIONAL along with STANDARD (Figures 3, 4). From Figure 3, we can see that CONDITIONAL typically improves coverage over the severely under-covered bins in Conf × Rank for all datasets. Figure 4 further demonstrates the effectiveness of our function class $\mathcal{F}$, as we see that coverage over all bins approaches the desired level of $1 - \alpha = 0.9$.

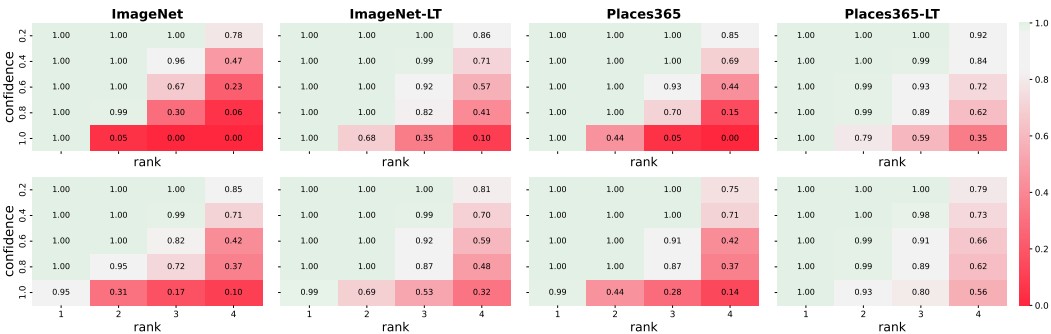

Figure 3: Conditional coverage of STANDARD (top) and CONDITIONAL (bottom) over regions of the feature space binned by model confidence and rank of the true class (Conf × Rank).

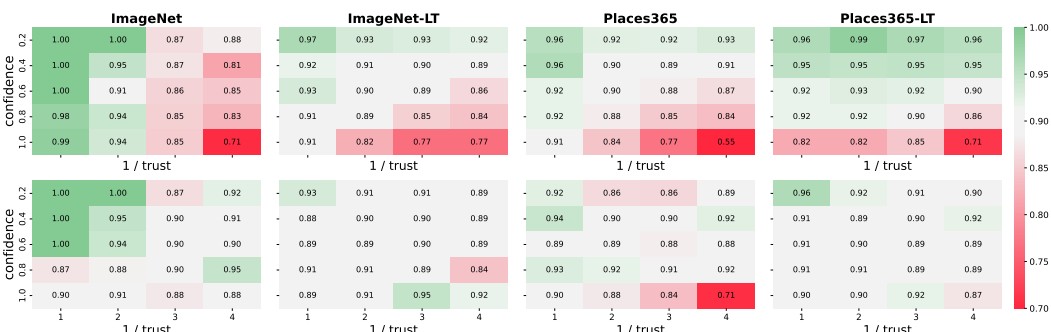

Figure 4: Conditional coverage of STANDARD (top) and CONDITIONAL (bottom) over regions of the feature space binned by model confidence and *trust score* (bottom) (Conf × Trust).

## C.2 EFFECT OF $d$

We study the effect of polynomial degree $d$ in function class $\mathcal{F}$ (14) on CovGap and average set size in Figure 5. We can see that the choice of $d$ is not a trivial one, and different datasets may have different optimal values for $d$. The polynomial function class offers greater flexibility in capturing interactions between the conformity score function and $V$ through this choice, compared to the NAIVE baseline. Specifically for ImageNet, it is interesting to note that the CovGap improves as we increase $d$, and the set sizes also shrink on average.

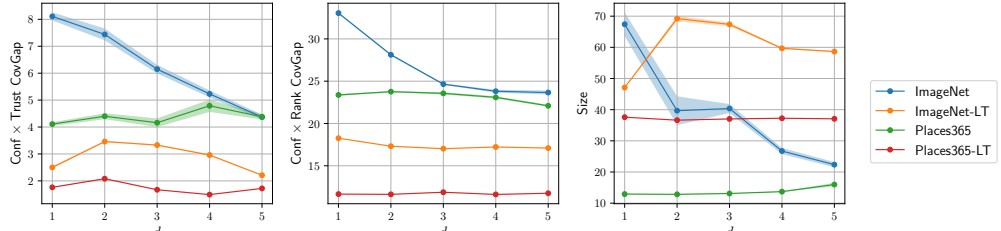

Figure 5: Effect of polynomial degree $d$ on CovGap and Size.

## C.3 TRUST-RANK CORRELATION

Here we empirically study the relationship between trust score (Trust) and rank of the true class (Rank). We compute the Pearson and Spearman's rank correlation coefficients with p-values for Trust and Rank on test samples in all datasets (Table 3). The correlation coefficients and p-values indicate a statistically significant negative correlation between Trust and Rank. We also plot the relationship between trust score and $\log(\text{rank})$ in Figure 6. This shows that lower (better) rank values generally correspond to higher trust scores on average, whereas higher (worse) rank values generally correspond to lower trust scores.

Table 3: Pearson and Spearman correlation coefficients with p-values for Trust and Rank.

| | **Pearson** | | **Spearman** | |
| --- | --- | --- | --- | --- |
| | **r** | **p-value** | **r** | **p-value** |
| **Dataset** | | | | |
| ImageNet | -0.09 | $< 0.001$ | -0.53 | $< 0.001$ |
| ImageNet-LT | -0.13 | $< 0.001$ | -0.47 | $< 0.001$ |
| Places365 | -0.02 | 0.004 | -0.43 | $< 0.001$ |
| Places365-LT | -0.14 | $< 0.001$ | -0.29 | $< 0.001$ |

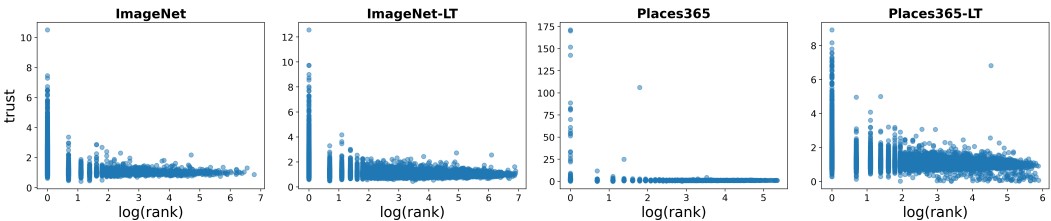

Figure 6: Relationship between trust score (Trust) and $\log(\text{rank})$ (Rank).

## C.4 PRINCIPAL COMPONENTS OF FEATURE LAYER AS $\mathcal{F}$

We construct an alternate function class using the top principal components of the feature layer. We choose the number of principal components as 20, with an added intercept term to achieve marginal coverage. We make this choice considering the computational constraints of the conditional conformal

procedure in Gibbs et al. (2023) at the scale of our datasets. We compare the performance of this function class with our method over all evaluation metrics. Approximate $X$-conditional coverage evaluation (Figure 7) shows that our method consistently outperforms this function class on all but one dataset. Evaluation metrics in Table 4 also show that CONDITIONAL outperforms this function class on average.

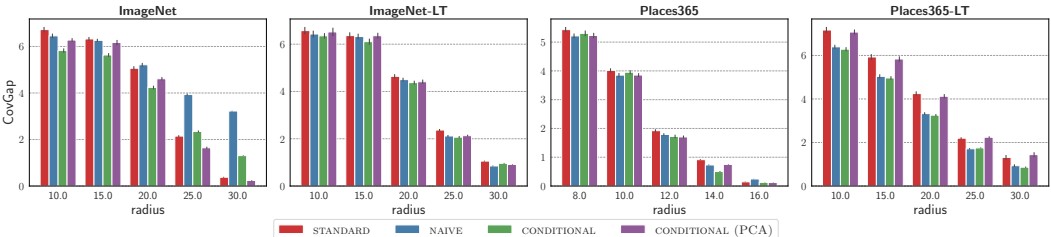

Figure 7: **Average coverage gap between randomly sampled Euclidean balls of fixed radius.** We vary the radius on x-axis. Standard errors are reported by error bars.

### C.5 COMPARISON WITH CLASSWISE MONDRIAN CP

We include the special case of Mondrian conformal predictio (CP) in our evaluation where each class forms a group (CLASSWISE). We split the calibration data by class and run conformal prediction once for each class. We evaluate the performance of this method over all evaluation metrics. In the approximate $X$-conditional coverage evaluation (Figure 8), we can see Mondrian CP performs worse than CONDITIONAL on all but one dataset. From conditional coverage evaluation in Table 5, we see that Mondrian CP achieves lower Conf × Rank CovGap on ImageNet-LT and Places365-LT, albeit with much larger set sizes. Interestingly, Mondrian CP does not always achieve the lowest Class-conditional coverage gap despite computing class-wise quantiles during calibration.

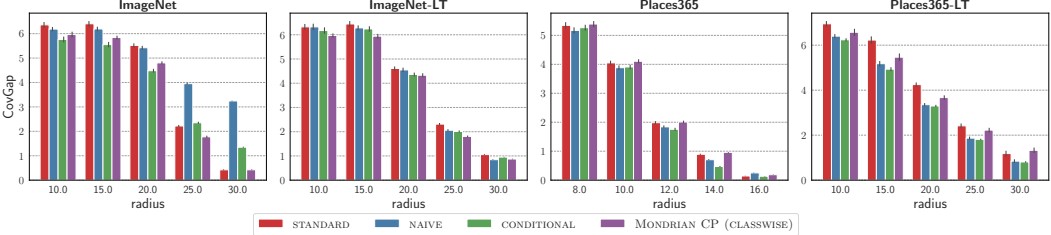

Figure 8: **Average coverage gap between randomly sampled Euclidean balls of fixed radius.** We vary the radius on x-axis. Standard errors are reported by error bars.

### C.6 CONDITIONAL COVERAGE EVALUATION USING DIFFERENT SCORE FUNCTIONS

We motivate our method using the *Adaptive Prediction Sets* (APS) algorithm (Romano et al., 2020b) designed to improve $X$-conditional coverage in classification settings. *Regularized Adaptive Prediction Sets* (RAPS) (Angelopoulos et al., 2021) is a regularized version of APS that produces smaller sets on average. In Table 6, we report the conditional coverage evaluation metrics on all datasets using APS, RAPS, and the simple softmax-based score described in Section 2.1. We note that our method can also improve conditional coverage properties using other score functions, especially in terms of class-conditional coverage gap.

Table 4: Conditional coverage evaluation on ImageNet, ImageNet-LT, Places365, and Places365-LT. Bold indicates the best (within ±0.1) coverage gap. We report standard errors in parentheses.

| Dataset | Method | Marginal | | Conditional | | |
| | | Coverage | Size | Conf × Trust CovGap | Conf × Rank CovGap | Class-conditional CovGap |
| --- | --- | --- | --- | --- | --- | --- |
| ImageNet | STANDARD | 0.90 (0.00) | 4.32 (0.04) | 6.35 (0.11) | 33.12 (0.15) | 7.23 (0.04) |
| | NAIVE | 0.93 (0.00) | 7.21 (0.05) | 8.01 (0.50) | 23.82 (0.15) | 6.68 (0.02) |
| | CONDITIONAL | 0.90 (0.00) | 22.36 (0.83) | **4.37** (0.09) | **23.66** (0.24) | **6.02** (0.04) |
| | CONDITIONAL (PCA) | 0.90 (0.00) | 6.54 (0.07) | 6.23 (0.09) | 31.27 (0.14) | 6.92 (0.04) |
| ImageNet-LT | STANDARD | 0.89 (0.00) | 50.35 (0.02) | 4.47 (0.02) | 19.92 (0.03) | 8.32 (0.00) |
| | NAIVE | 0.89 (0.00) | 46.63 (0.07) | 2.99 (0.01) | 18.27 (0.02) | 8.29 (0.01) |
| | CONDITIONAL | 0.90 (0.00) | 58.64 (0.18) | **2.21** (0.01) | **17.09** (0.02) | **7.92** (0.00) |
| | CONDITIONAL (PCA) | 0.89 (0.00) | 53.59 (0.05) | 4.78 (0.01) | 20.13 (0.04) | 8.17 (0.01) |
| Places365 | STANDARD | 0.90 (0.00) | 14.17 (0.07) | 5.78 (0.08) | 25.58 (0.09) | **4.99** (0.05) |
| | NAIVE | 0.90 (0.00) | 12.76 (0.11) | **2.40** (0.16) | 22.22 (0.13) | 5.11 (0.06) |
| | CONDITIONAL | 0.90 (0.00) | 15.98 (0.57) | 4.38 (0.10) | **22.09** (0.12) | **4.98** (0.05) |
| | CONDITIONAL (PCA) | 0.90 (0.00) | 13.89 (0.07) | 5.34 (0.07) | 25.17 (0.12) | **4.96** (0.04) |
| Places365-LT | STANDARD | 0.90 (0.00) | 43.46 (0.10) | 5.55 (0.02) | 13.23 (0.01) | **5.34** (0.00) |
| | NAIVE | 0.90 (0.00) | 38.54 (0.06) | 4.55 (0.02) | **11.75** (0.01) | 5.61 (0.01) |
| | CONDITIONAL | 0.90 (0.00) | 37.07 (0.03) | **1.72** (0.00) | **11.75** (0.01) | 5.65 (0.00) |
| | CONDITIONAL (PCA) | 0.90 (0.00) | 45.03 (0.13) | 5.37 (0.03) | 12.89 (0.01) | **5.35** (0.01) |

Table 5: Conditional coverage evaluation on ImageNet, ImageNet-LT, Places365, and Places365-LT. Bold indicates the best (within ±0.1) coverage gap. We report standard errors in parentheses.

| Dataset | Method | Marginal | | Conditional | | |
| | | Coverage | Size | Conf × Trust CovGap | Conf × Rank CovGap | Class-conditional CovGap |
| --- | --- | --- | --- | --- | --- | --- |
| ImageNet | STANDARD | 0.90 (0.00) | 4.32 (0.04) | 6.35 (0.11) | 33.12 (0.15) | 7.23 (0.04) |
| | NAIVE | 0.93 (0.00) | 7.21 (0.05) | 8.01 (0.50) | 23.82 (0.15) | 6.68 (0.02) |
| | CONDITIONAL | 0.90 (0.00) | 22.36 (0.83) | **4.37** (0.09) | **23.66** (0.24) | **6.02** (0.04) |
| | MONDRIAN CP | 0.90 (0.00) | 9.83 (0.14) | 7.83 (0.32) | 26.70 (0.16) | 6.36 (0.03) |
| ImageNet-LT | STANDARD | 0.89 (0.00) | 50.35 (0.02) | 4.47 (0.02) | 19.92 (0.03) | 8.32 (0.00) |
| | NAIVE | 0.89 (0.00) | 46.63 (0.07) | 2.99 (0.01) | 18.27 (0.02) | 8.29 (0.01) |
| | CONDITIONAL | 0.90 (0.00) | 58.64 (0.18) | **2.21** (0.01) | 17.09 (0.02) | 7.92 (0.00) |
| | MONDRIAN CP | 0.90 (0.00) | 67.38 (0.10) | 3.77 (0.01) | **15.23** (0.02) | **6.34** (0.00) |
| Places365 | STANDARD | 0.90 (0.00) | 14.17 (0.07) | 5.78 (0.08) | 25.58 (0.09) | 4.99 (0.05) |
| | NAIVE | 0.90 (0.00) | 12.76 (0.11) | **2.40** (0.16) | 22.22 (0.13) | 5.11 (0.06) |
| | CONDITIONAL | 0.90 (0.00) | 15.98 (0.57) | 4.38 (0.10) | **22.09** (0.12) | 4.98 (0.05) |
| | MONDRIAN CP | 0.90 (0.00) | 17.63 (0.08) | 5.09 (0.07) | 22.79 (0.17) | **4.68** (0.05) |
| Places365-LT | STANDARD | 0.90 (0.00) | 43.46 (0.10) | 5.55 (0.02) | 13.23 (0.01) | **5.34** (0.00) |
| | NAIVE | 0.90 (0.00) | 38.54 (0.06) | 4.55 (0.02) | 11.75 (0.01) | 5.61 (0.01) |
| | CONDITIONAL | 0.90 (0.00) | 37.07 (0.03) | **1.72** (0.00) | 11.75 (0.01) | 5.65 (0.00) |
| | MONDRIAN CP | 0.90 (0.00) | 60.34 (0.13) | 5.12 (0.02) | **11.27** (0.02) | 5.47 (0.01) |

Table 6: Conditional coverage evaluation on ImageNet, ImageNet-LT, Places365, and Places365-LT with different conformity score functions. Bold indicates the best (within ±0.1) coverage gap. We report standard errors in parentheses.

| | | | Marginal | | Conditional | | |
| | | | Coverage | Size | Conf × Trust CovGap | Conf × Rank CovGap | Class-conditional CovGap |
| **Dataset** | **Score func.** | **Method** | | | | | |
| ImageNet | softmax | STANDARD | 0.90 (0.00) | 2.12 (0.01) | 10.32 (0.15) | 32.25 (0.12) | 7.41 (0.04) |
| | | NAIVE | 0.90 (0.00) | 4.61 (0.06) | 8.48 (0.62) | 24.18 (0.12) | 6.28 (0.04) |
| | | CONDITIONAL | 0.90 (0.00) | 98.36 (1.84) | **5.94** (0.26) | **23.89** (0.17) | **6.09** (0.04) |
| | APS | STANDARD | 0.90 (0.00) | 4.32 (0.04) | 6.35 (0.11) | 33.12 (0.15) | 7.23 (0.04) |
| | | NAIVE | 0.93 (0.00) | 7.21 (0.05) | 8.01 (0.50) | 23.82 (0.15) | 6.68 (0.02) |
| | | CONDITIONAL | 0.90 (0.00) | 22.36 (0.83) | **4.37** (0.09) | **23.66** (0.24) | **6.02** (0.04) |
| | RAPS | STANDARD | 0.90 (0.08) | 3.03 (0.02) | 7.03 (0.08) | 33.12 (0.18) | 7.27 (0.04) |
| | | NAIVE | 0.93 (0.00) | 5.13 (0.04) | 7.99 (0.50) | 24.04 (0.15) | 6.73 (0.01) |
| | | CONDITIONAL | 0.90 (0.00) | 5.82 (0.06) | **4.45** (0.19) | **23.76** (0.23) | **6.07** (0.02) |
| ImageNet-LT | softmax | STANDARD | 0.90 (0.00) | 24.54 (0.02) | 4.25 (0.02) | **17.68** (0.02) | 9.45 (0.01) |
| | | NAIVE | 0.89 (0.00) | 33.04 (0.04) | **2.86** (0.00) | 19.86 (0.01) | 8.37 (0.00) |
| | | CONDITIONAL | 0.90 (0.00) | 228.82 (1.44) | 6.82 (0.01) | 18.04 (0.01) | **7.68** (0.01) |
| | APS | STANDARD | 0.89 (0.00) | 50.35 (0.02) | 4.47 (0.02) | 19.92 (0.03) | 8.32 (0.00) |
| | | NAIVE | 0.89 (0.00) | 46.63 (0.07) | 2.99 (0.01) | 18.27 (0.02) | 8.29 (0.01) |
| | | CONDITIONAL | 0.90 (0.00) | 58.64 (0.18) | **2.21** (0.01) | **17.09** (0.02) | **7.92** (0.00) |
| | RAPS | STANDARD | 0.89 (0.00) | 25.79 (0.08) | 5.30 (0.02) | **16.95** (0.03) | 9.55 (0.00) |
| | | NAIVE | 0.89 (0.00) | 33.82 (0.00) | 2.98 (0.01) | 19.95 (0.02) | 8.50 (0.01) |
| | | CONDITIONAL | 0.90 (0.00) | 35.04 (0.02) | **1.72** (0.01) | 18.73 (0.02) | **8.15** (0.01) |
| Places365 | softmax | STANDARD | 0.90 (0.00) | 9.38 (0.03) | 3.18 (0.06) | **20.21** (0.13) | 5.30 (0.05) |
| | | NAIVE | 0.90 (0.00) | 10.37 (0.06) | **2.39** (0.14) | 22.36 (0.10) | 5.04 (0.05) |
| | | CONDITIONAL | 0.90 (0.00) | 43.96 (1.68) | 6.64 (0.21) | 23.15 (0.12) | **4.86** (0.04) |
| | APS | STANDARD | 0.90 (0.00) | 14.17 (0.07) | 5.78 (0.08) | 25.58 (0.09) | **4.99** (0.05) |
| | | NAIVE | 0.90 (0.00) | 12.76 (0.11) | **2.40** (0.16) | 22.22 (0.13) | 5.11 (0.06) |
| | | CONDITIONAL | 0.90 (0.00) | 15.98 (0.57) | 4.38 (0.10) | **22.09** (0.12) | **4.98** (0.05) |
| | RAPS | STANDARD | 0.90 (0.00) | 10.24 (0.05) | 3.60 (0.08) | **21.43** (0.13) | 5.21 (0.04) |
| | | NAIVE | 0.90 (0.00) | 10.93 (0.09) | **2.44** (0.14) | 22.38 (0.09) | 5.05 (0.05) |
| | | CONDITIONAL | 0.90 (0.00) | 11.48 (0.05) | 3.99 (0.09) | 22.19 (0.10) | **4.99** (0.04) |
| Places365-LT | softmax | STANDARD | 0.90 (0.00) | 24.98 (0.05) | 2.59 (0.00) | 14.88 (0.01) | **6.24** (0.00) |
| | | NAIVE | 0.90 (0.00) | 26.45 (0.03) | 4.62 (0.01) | 14.98 (0.02) | 6.36 (0.01) |
| | | CONDITIONAL | 0.90 (0.00) | 28.35 (0.25) | **2.02** (0.01) | **14.44** (0.01) | **6.23** (0.01) |
| | APS | STANDARD | 0.90 (0.00) | 43.46 (0.10) | 5.55 (0.02) | 13.23 (0.01) | **5.34** (0.00) |
| | | NAIVE | 0.90 (0.00) | 38.54 (0.06) | 4.55 (0.02) | **11.75** (0.01) | 5.61 (0.01) |
| | | CONDITIONAL | 0.90 (0.00) | 37.07 (0.03) | **1.72** (0.00) | **11.75** (0.01) | 5.65 (0.00) |
| | RAPS | STANDARD | 0.90 (0.00) | 23.46 (0.10) | 3.00 (0.01) | 15.84 (0.11) | 6.47 (0.00) |
| | | NAIVE | 0.90 (0.00) | 26.23 (0.04) | 4.31 (0.01) | 14.91 (0.05) | 6.44 (0.01) |
| | | CONDITIONAL | 0.90 (0.00) | 25.92 (0.05) | **1.51** (0.01) | **14.81** (0.05) | **6.37** (0.01) |

