# OpenReview forum: "Conformal Prediction Sets with Improved Conditional Coverage using Trust Scores"
_ICLR.cc/2025/Conference — Submitted to ICLR 2025_

### Official Review · Reviewer_YaHX · 2024-11-02

**Soundness:** 3
**Presentation:** 3
**Contribution:** 2
**Rating:** 5
**Confidence:** 4

**Summary:**

This paper focuses on a fundamental limitation in conformal prediction: while providing marginal coverage guarantees, it lacks conditional coverage properties critical for reliable uncertainty quantification. The authors propose a novel approach that leverages model confidence and trust scores to achieve approximate conditional coverage, particularly targeting instances where classifiers exhibit overconfidence in misclassifications. Their method demonstrates superior empirical performance on large-scale vision datasets while maintaining theoretical guarantees, advancing the practical applicability of conformal prediction in high-stakes applications.

**Strengths:**

- The paper's motivation is clear and the writing is well-organized, making it easy to follow.
- The extensive reproducibility of the experiment is greatly appreciated.
- The proposed method improves conditional coverage significantly, though at the cost of larger prediction set sizes.

**Weaknesses:**

- The Proposition 1 is strict. As shown in  Proposition 1, higher temperature has larger prediction sets. However, this seems inconsistent with prior works [1] where the relationship between temperature and set size is not monotonic. Moreover, Proposition 1 implies that as temperature approaches infinity, the set size would converge to the total number of classes. Could the authors provide a more comprehensive discussion on how different temperature values (both small and large) affect the method's performance?
- The experiments are limited. While the proposed method achieves better conditional coverage, it comes at the cost of larger prediction set sizes. For a fair comparison, the authors should include results from relevant baselines such as Cluster CP [2] and Mondrian CP [3]. Additionally, experiments on datasets with fewer classes (like CIFAR10 and CIFAR100) would be valuable, especially since Mondrian CP has demonstrated strong class-conditional coverage performance on these datasets. Next, the ablation analysis about k of k-NN radius in trust score computation would be helpful.

[1] Cha, Seohyeon, Honggu Kang, and Joonhyuk Kang. "On the temperature of bayesian graph neural networks for conformal prediction."
[2] Tiffany Ding, Anastasios Nikolas Angelopoulos, Stephen Bates, Michael Jordan, and Ryan Tibshirani. Class-conditional conformal prediction with many classes.
[3] Vladimir Vovk, David Lindsay, Ilia Nouretdinov, and Alex Gammerman. Mondrian confidence machine.

**Questions:**

- Are authors using the non-randomized version of RAPS in Appendix C.4, similar to how authors implemented APS?
- For the long-tailed datasets (ImageNet-LT and Places365-LT), do the test sets maintain the same imbalanced distribution? If so, this could lead to unreliable evaluation metrics for minority classes due to insufficient test samples. Have authors considered this potential bias in evaluation?
- Since the trust score is computed using features from the model's penultimate layer, could we explore using alternative encoders for trust score calculation? A better encoder might provide more discriminative features, potentially leading to more accurate trust scores and consequently smaller prediction sets.

---

> ### Author Response · Authors · 2024-11-22
> **Official Response by Authors [1/2]**
>
> Thank you for your review, and in particular, for highlighting the ability of our method to advance the practical applicability of conformal prediction in high-stakes applications. We respond to specific questions and remarks below.
>
> > Regarding Proposition 1
>
> Thank you for bringing this point to our notice. Our result holds conditionally on $\hat{q}$, and not marginally. We have updated the paper to clarify this. The paper cited by the reviewer studies the interactive effects between temperature, the calibration scheme, and the set size, whereas our result only studies the effect of temperature, holding everything else fixed and is meant to provide insight into why the patterns in Figure 1 emerge. Our goal is to show that if the true $Y$ is ranked further down in the sorted order of probabilities and the model is more confident, the probability of miscoverage is higher than if it was less confident. The temperature allows us to tune the amount of confidence that goes to the top score in a parametric way. To answer your question, temperature does not interact with our method in a unique way and has the same effect as shown in prior work. If we have similar assumptions as in prior works, we will achieve a similar non-monotonic relationship.
>
> > Regarding class-conditional coverage and inclusion of baselines
>
> That is true, our method outputs sets that are slightly larger than standard conformal prediction on average, however this is not surprising given the consistent improvement in conditional coverage – there is no free lunch!
>
> Regarding the reviewer’s comment on class-conditional coverage, we would like to emphasize that *our primary goal is not to achieve class-conditional coverage* as it is in the methods mentioned above; rather, we aim to improve coverage over instances where the classifier is overconfident in its incorrect predictions as identified by our lower-dimensional statistic. We evaluate over a suite of metrics to show our method *generally* improves conditional coverage as a consequence, including class-conditional coverage over standard conformal prediction. We include a special case of Mondrian CP as we mention in Section 4.3, 4.4 in the form of NAIVE baseline, where the procedure has access to group information it will be evaluated on.
>
> That said, methods for class-conditional CP can potentially improve approximate $X$-conditional coverage if the class-wise computed quantiles align with quantiles of the conditional distribution, however this is not always expected or guaranteed. Following the reviewer’s suggestion, we perform an evaluation with Mondrian CP that splits the calibration data by class and runs conformal prediction once for each class. We add the results table below and include a new figure with approximate $X$-conditional coverage evaluation in the paper (Appendix C.5, Figure 8). We see that Mondrian CP improves $\textsf{Conf} \times \textsf{Rank}$ coverage gap in ImageNet-LT and Places365-LT at the cost of **even larger set sizes**. In the approximate $X$-conditional coverage evaluation, we see Mondrian CP performs worse than our Conditional method. Interestingly, Mondrian CP does not always achieve the lowest Class-conditional coverage gap despite computing class-wise quantiles during calibration.

---

> ### Author Response · Authors · 2024-11-22
> **Official Response by Authors [2/2]**
>
> | Dataset | Method | Marginal |  | Conditional |  |  |
> |---|---|---|---|---|---|---|
> |  |  | Coverage | Size | Conf x Trust CovGap | Conf x Rank CovGap | Class-conditional CovGap |
> | ImageNet | STANDARD | 0.90 (0.00) | 4.32 (0.04) | 6.35 (0.11) | 33.12 (0.15) | 7.23 (0.04) |
> |  | NAIVE | 0.93 (0.00) | 7.21 (0.05) | 8.01 (0.50) | 23.82 (0.15) | 6.68 (0.02) |
> |  | CONDITIONAL | 0.90 (0.00) | 22.36 (0.83) | **4.37** (0.09) | **23.66** (0.24) | **6.02** (0.04) |
> |  | MONDRIAN CP (CLASSWISE) | 0.90 (0.00) | 9.83 (0.14) | 7.83 (0.32) | 26.70 (0.16) | 6.36 (0.03) |
> | ImageNet-LT | STANDARD     | 0.89 (0.00) | 50.35 (0.02)| 4.47 (0.02) | 19.92 (0.03) | 8.32 (0.00) |
> |                 | NAIVE        | 0.89 (0.00) | 46.63 (0.07)| 2.99 (0.01) | 18.27 (0.02) | 8.29 (0.01) |
> |                 | CONDITIONAL  | 0.90 (0.00) | 58.64 (0.18)| **2.21** (0.01) | 17.09 (0.02) | 7.92 (0.00) |
> |  | MONDRIAN CP (CLASSWISE) | 0.90 (0.00) | 67.38 (0.10) | 3.77 (0.01) | **15.23** (0.02) | **6.34** (0.00) |
> | Places365   | STANDARD     | 0.90 (0.00) | 14.17 (0.07)| 5.78 (0.08) | 25.58 (0.09) | 4.99 (0.05) |
> |                 | NAIVE        | 0.90 (0.00) | 12.76 (0.11)| **2.40** (0.16) | 22.22 (0.13) | 5.11 (0.06) |
> |                 | CONDITIONAL  | 0.90 (0.00) | 15.98 (0.57)| 4.38 (0.10) | **22.09** (0.12) | 4.98 (0.05) |
> |  | MONDRIAN CP (CLASSWISE) | 0.90 (0.00) | 17.63 (0.08) | 5.09 (0.07) | 22.79 (0.17) | **4.68** (0.05) |
> | Places365-LT| STANDARD     | 0.90 (0.00) | 43.46 (0.10)| 5.55 (0.02) | 13.23 (0.01) | **5.34** (0.00) |
> |                 | NAIVE        | 0.90 (0.00) | 38.54 (0.06)| 4.55 (0.02) | 11.75 (0.01) | 5.61 (0.01) |
> |                 | CONDITIONAL  | 0.90 (0.00) | 37.07 (0.03)| **1.72** (0.00) | 11.75 (0.01) | 5.65 (0.00) |
> |  | MONDRIAN CP (CLASSWISE) | 0.90 (0.00) | 60.34 (0.13) | 5.12 (0.02) | **11.27** (0.02)  | 5.47 (0.00) |
>
> > Ablation on $k$ of k-NN radius in trust score computation
>
> As we mentioned in Appendix B.3, we follow the original work (Jiang et al., 2018 [1]) to skip the initial filtering step in the trust score implementation for improving computational efficiency. The $k$-nearest neighbor calculations make up the bulk of the computational cost, and the original work found the procedure to not be very sensitive to the choice of $k$. Our datasets are larger than the ones used in [1], and hence this choice made intuitive sense. This also leads to a hyperparameter-free procedure as noted by Jiang et al.
>
> In response to questions:
>
> > Are authors using the non-randomized version of RAPS in Appendix C.4, similar to how authors implemented APS?
>
> Yes, we are consistent in using the non-randomized version in order to generate deterministic prediction sets and maintain fair comparison.
>
> > For the long-tailed datasets (ImageNet-LT and Places365-LT), do the test sets maintain the same imbalanced distribution? If so, this could lead to unreliable evaluation metrics for minority classes due to insufficient test samples. Have authors considered this potential bias in evaluation?
>
> The reviewer raises a valid point. We use ImageNet-LT and Places365-LT as introduced by [2], where the training set is imbalanced while the validation and test sets are balanced. As mentioned in the appendix, we obtain data splits from previous benchmarks [3, 4].
>
> > Since the trust score is computed using features from the model's penultimate layer, could we explore using alternative encoders for trust score calculation? A better encoder might provide more discriminative features, potentially leading to more accurate trust scores and consequently smaller prediction sets.
>
> That is an interesting direction to explore, and there can be variants that improve performance. We chose a representation that made intuitive sense and found it to work well in practice. Better encoders can lead to better approximation using trust scores and improve performance even further. We will highlight this in the paper.
>
> [1] Jiang, H., Kim, B., & Gupta, M.R. (2018). To Trust Or Not To Trust A Classifier. Neural Information Processing Systems.
>
> [2] Liu, Z., Miao, Z., Zhan, X., Wang, J., Gong, B., & Yu, S.X. (2019). Large-Scale Long-Tailed Recognition in an Open World. 2019 IEEE/CVF Conference on Computer Vision and Pattern Recognition (CVPR), 2532-2541.
>
> [3] Zhong, Z., Cui, J., Liu, S., & Jia, J. (2020). Improving Calibration for Long-Tailed Recognition. 2021 IEEE/CVF Conference on Computer Vision and Pattern Recognition (CVPR), 16484-16493.
>
> [4] Yuksekgonul, M., Zhang, L., Zou, J.Y., & Guestrin, C. (2023). Beyond Confidence: Reliable Models Should Also Consider Atypicality. ArXiv, abs/2305.18262.

---

> ### Author Response · Authors · 2024-11-29
> **Requesting Review of Author Response**
>
> Thank you again for taking the time to review our paper. As we approach the end of the discussion phase, we would like to confirm if our responses addressed your comments. In case there are any remaining questions or clarifications required, we are happy to discuss further.
>
> Thank you,
>
> Authors

---

> > ### Comment · Reviewer_YaHX · 2024-12-02
> >
> > Thanks for the detailed response.
> > > Regarding Proposition 1
> >
> > The proposed constraint on $\hat{q}$ may not reflect realistic conditions. The value of $\hat{q}$ should vary with temperature. And, my experimental results show that at high temperatures, the average set size decreases as temperature increases. Perhaps the authors could consider making Proposition 1 clearer, or they might want to consider removing it.
> >
> > Therefore, I maintain my score.

---

> ### Author Response · Authors · 2024-12-02
> **Official Comment by Authors**
>
> We are glad we were able to address the rest of your concerns. We understand your point here, and we will surely make Proposition 1 clearer in the final version and contextualize it with respect to our specific setting, which as we explained in our rebuttal, differs from the setting in the paper you cited. Also, we are not sure what you refer to as *"my experimental results"*, so we would perhaps not be able to comment on that.
>
> That said, the above point is secondary to our main contribution which is largely appreciated by you as well as other reviewers. We hope you reconsider whether the significance of our contribution is appropriately weighted in your final assessment. We are happy to continue the discussion and incorporate any further suggestions you have.

---

> > ### Comment · Reviewer_YaHX · 2024-12-02
> >
> > Sorry for the misunderstanding. I conducted the temperature experiment from paper [1] myself. I found that when the temperature exceeds a certain value, the higher the temperature, the smaller the prediction set becomes. Therefore, the statement in line 211 "overconfident models concentrate the probability mass on fewer classes, yielding smaller prediction sets. " should be revised. Overconfident predictions may actually have long-tailed probabilities, which can lead to larger prediction sets for APS.
> >
> > For proposition 1, can we compare the coverage probability of two samples under a fixed temperature? This will be the same as the results of Figure 1. Otherwise, I don't understand what the point of comparison at different temperatures is, and Proposition 1 doesn't seem to have much to do with your whole work.
> >
> > [1] Cha, Seohyeon, Honggu Kang, and Joonhyuk Kang. "On the temperature of bayesian graph neural networks for conformal prediction."

---

> ### Author Response · Authors · 2024-12-02
> **Official Comment by Authors**
>
> Thank you for your continued engagement.
>
> Unfortunately, we cannot compare the coverage probability for two samples as the model probabilities beyond the top output may be arbitrarily different. We did consider this initially, however it is not possible to marginally explain the coverage results in Figure 1 without further assumptions.
>
> Please note that the goal of this proposition is not to study the effect of temperature but rather use the temperature as a way to parameterize confidence to study the relation between confidence and miscoverage in a stylized setting. The parameterization allows us to vary the amount of confidence that goes to the top score while keeping everything else fixed, and hence we study conditional miscoverage in Proposition 1. The result in this proposition is meant to be illustrative and is not central to our contribution or proposed methods. We will explain this more carefully in the camera-ready version and include the clarification suggested by the reviewer. If the reviewer feels it might cause some confusion to readers or muddle our main contributions, we can consider moving it.
>
> We hope this resolves your concern, and we are happy to discuss further.

---

### Official Review · Reviewer_xou9 · 2024-11-03

**Soundness:** 2
**Presentation:** 3
**Contribution:** 2
**Rating:** 3
**Confidence:** 4

**Summary:**

This paper introduces a conformal prediction approach for classification aimed at improving conditional coverage properties. While achieving finite-sample X-conditional coverage guarantees is nearly impossible, the authors propose a method that provides V-conditional coverage guarantees, where V is a simplified representation of X. This representation partitions the instance space into regions based on model confidence (mode of the predicted probability distribution) and the rank of the true label. Since the rank of the true label is unknown for unobserved test instances, the authors use a reliability or trust score to approximate the model’s performance relative to the Bayes predictor.

**Strengths:**

The paper is well-written, easy to follow, and addresses an important challenge in conformal prediction: the lack of conditional coverage. The proposed method is theoretically sound and practically applicable, as demonstrated by the experimental results.

**Weaknesses:**

1. Although the trust scores were intended to be a proxy of rank function, experimental results—comparing the coverage gaps between the Conf×Trust and Conf×Rank columns in all tables—indicate this was not achieved. Additionally, the proposed method’s improvement over the Naive approach in terms of the Conf×Rank coverage gap is consistently negligible, suggesting that the method may not be particularly effective.

2. The comparison of the average set size between the Naive approach and the proposed method suggests that the proposed approach becomes overly conservative in many cases while offering only a minimal benefit in terms of the desired conditional coverage.


3. Even though the current evaluation metrics in Section 4 are interesting, given the paper's aim to improve X-conditional coverage, it would have been helpful to include additional metrics used in other studies, such as WSC from the APS paper or SSCV from the RAPS paper.

**Questions:**

The proposed approach requires discretization of the space V. Have the authors explored the effect of different discretization methods?

---

> ### Author Response · Authors · 2024-11-22
> **Official Response by Authors [1/2]**
>
> Thank you for your review, and for acknowledging the importance of the problem we address through our work. We are glad you found our approach to be theoretically sound and practically applicable, and our paper easy to follow.
>
> In response to weaknesses:
>
> > W1: Regarding Conf×Trust and Conf×Rank coverage gaps
>
> Our aim was to identify a lower dimensional statistic indicative of instances where conformal prediction is most likely to fail—specifically when the classifier is *overconfident in its incorrect predictions*. As can be seen in Figure 1, trust score approximately captures the *patterns of miscoverage* as the rank-based partitions. However, the exact values of coverage are not expected to be similar — rank can be thought of as an oracle metric that neatly partitions the space into bins ranging from coverage of 0 to 1. The trust scores are used as a proxy to identify test instances prone to error, enabling us to improve coverage along the Conf×Trust and indirectly Conf×Rank axes. We did not expect, and hopefully did not claim that the coverage gap values will be close. We are happy to clarify this in the final version if needed.
>
> > W1, W2: Comparison between Naive and Conditional
>
> We would like to start by **emphasizing that Naive might have been a misnomer since it is an ablation of our method rather than an existing or a naively implemented procedure.** We understand that this terminology might be confusing, and will make sure to update this in the final version but will refer to it as such in the rebuttal. Naive runs conformal prediction within each ConfxTrust bin in Table 1 (and Figure 2) and within each skin type group in Table 2 – these are the same groups that are *also used in evaluation*. That is to say, **Naive has strictly more information about the evaluation groups,** which makes it a very competitive baseline.
> In what follows, we discuss the comparative results of Naive and Conditional in Section 4.3 and Section 4.4 individually:
>
> 1. **Section 4.3**: The success of both Naive and Conditional method over standard conformal prediction in Table 1 indicate that *our proposed lower-dimensional statistic $V$ is effective* in improving conditional coverage, since both methods use our proposed statistic $V$. While Naive has access to the discrete binning we use for evaluation, our proposed Conditional method still outperforms Naive in terms of the ConfxRank coverage gap, which demonstrates the utility of our proposed function class $\mathcal{F}$.
>
>
>      Additionally, the Naive approach has conceptual limitations—the discretization involved in this baseline may limit its generalization to other forms of conditional coverage, as we can see from the approximate $X$-conditional coverage evaluation. Moreover, the success of the Naive approach depends on whether there are adequate numbers of samples in each bin to compute a non-trivial quantile. Our proposed approach faces no such constraint.
>
> 2. **Section 4.4**: Our evaluation on Fitzpatrick 17k also highlights the effectiveness of our method. Here, we consider an analogous Groupwise baseline to Naive. Despite Groupwise having access to skin type group labels during calibration, our method outperforms the approach in terms of all metrics. This comes with no significant increase in size. This demonstrates the utility of our method in real-world applications, especially when access to group labels is limited or collection may be expensive.
>
> > W3: Regarding inclusion of additional metrics
>
> Thank you for your suggestion. We perform evaluation with the SSCV metric for all datasets and methods. We would like to note that the RAPS paper does not provide a clear protocol to select the set-size strata. For our evaluation, we use 6 partitions and choose the strata based on set-size quantiles of standard conformal prediction, ensuring that the evaluation is not biased/favorable toward our method. Despite that, we find Conditional and Naive methods to outperform standard conformal prediction in terms of this metric as well.
>
> | Dataset | Method | SSCV |
> |---------------|------------|----------------------|
> | ImageNet | STANDARD | 0.088	(0.003) |
> | | NAIVE | **0.055**	(0.001) |
> | | CONDITIONAL | 0.080	(0.001) |
> | ImageNet-LT| STANDARD | 0.046	(0.000) |
> | | NAIVE | 0.036	(0.000) |
> | | CONDITIONAL | **0.028**	(0.000) |
> | Places365 | STANDARD | 0.038	(0.001) |
> | | NAIVE | **0.026**	(0.002) |
> | | CONDITIONAL | 0.032	(0.001) |
> | Places365-LT | STANDARD | 0.136	(0.000) |
> | | NAIVE | 0.086	(0.000) |
> | | CONDITIONAL | **0.077**	(0.000) |

---

> > ### Author Response · Authors · 2024-11-22
> > **Official Response by Authors [2/2]**
> >
> > In response to questions:
> >
> > >  The proposed approach requires discretization of the space V. Have the authors explored the effect of different discretization methods?
> >
> > We believe there is some misunderstanding here  – as we discussed above, the advantage of our method is that we *do not* require discretization of the space. We did not discretize the space $V$ and modeled a bivariate function of two continuous variables. We are happy to further clarify this in the paper based on the reviewer's suggestions.

---

> > ### Comment · Reviewer_xou9 · 2024-11-26
> >
> > Thank you for elaborating on my comments.
> >
> > In the paper, you begin reasoning about your proposed approach by highlighting that Conf×Rank is what matters and then proceed to suggest using the trust score as a proxy since Rank is not available. In the rebuttal, by referring to Figure 1 again, you seem to imply that the goal is for Conf×Trust to approximately capture the same patterns as Conf×Rank. One way to assess the quality and usefulness of this approximation is by evaluating whether the conformal results for Conf×Trust align closely with the oracle results of Conf×Rank, which, based on your findings, does not seem to be the case. Otherwise, you could omit the Conf×Rank discussion entirely and directly present Conf×Trust as the chosen conditioning factor. So, I believe my critique here is fair.
> >
> > Regarding the naive approach, my point is that if your contribution were solely to define a Mondrian approach based on a new criterion—which you correctly label as naive—then your contribution would be highly incremental. In that case, you should have compared it with many other existing Mondrian criteria to demonstrate whether your approach provides any tangible benefit.
> > However, if we consider the main contribution to be the conditional approach, my critique remains valid: it tends to be overly conservative in many cases while offering only minimal improvement in achieving the desired conditional coverage. I completely agree with the author regarding the limitations of the naive approach. However, the conditional approach is also not without its limitations, such as the added computational complexity.

---

> ### Author Response · Authors · 2024-11-30
> **Official Comment by Authors [1/2]**
>
> Thank you for engaging with us and for your comments. We respond pointwise below:
>
> > In the paper, you begin reasoning about your proposed approach by highlighting that Conf×Rank is what matters and then proceed to suggest using the trust score as a proxy since Rank is not available... Otherwise, you could omit the Conf×Rank discussion entirely and directly present Conf×Trust as the chosen conditioning factor. So, I believe my critique here is fair.
>
> The $\textsf{Rank}$ variable, by definition, inherently captures miscoverage patterns, as miscoverage events are characterized by a high $\textsf{Rank}$ for the true label and low confidence—a relationship we empirically validate in Figure 1. Our goal is to **identify and improve coverage over instances where a classifier is overconfident in its incorrect predictions**, as identified by a lower dimensional statistic $V$. The $\textsf{Trust}$ score is intended to approximate the same miscoverage trend as $\textsf{Rank}$, and not the exact value of $\textsf{Rank}$. In line with our goal above, the $\textsf{Trust}$ score can help identify instances where the classifier is incorrectly over-confident. So, bins based on ConfxTrust may not be identical to ConfxRank and that is not surprising.
>
> The justification for the use of $\textsf{Trust}$ score for this approximation as follows: Theorem 4 in Jiang et al., 2018 demonstrates that, under certain regularity assumptions, the $\textsf{Trust}$ score can indicate whether the model $f(X)$ aligns with the Bayes-Optimal model $f^*(X)$ as:
>
> $\textsf{Trust}(X) < \gamma \Rightarrow f(X) \neq f^*(X),$
>
> $1/\textsf{Trust}(X) < \gamma \Rightarrow f(X) = f^*(X),$
>
> for some threshold $\gamma$. This result suggests that the $\textsf{Trust}$ score stratifies prediction instances based on their alignment with the Bayes-Optimal classifier. In essence, it provides a coarser analog to the binning in Figure 1 into $\textsf{Rank}=1$ and $\textsf{Rank}>1$ categories. Since the $\textsf{Trust}$ score evaluates alignment with the Bayes-Optimal classifier through thresholding, it is intuitive that higher continuous values of the $\textsf{Trust}$ score correlate with increasing $\textsf{Rank}$ scores.
>
> We evaluate the *quality and usefulness of this approximation* by evaluating the **improvement in conditional coverage over standard conformal prediction** (i.e., reduction in coverage gap (Table 1, Table 2, Figure 2, Figure 3)). Figure 3 specifically shows that we are able to improve coverage in instances where a classifier is overconfident in its incorrect predictions. ConfxRank coverage gap (CovGap) is one of the metrics we evaluate on and improve coverage on; however, this notion improves conditional coverage generally as evaluated by approximate $X$-conditional coverage (Figure 2), coverage over demographic subgroups (Table 2), and class-conditional coverage (Figure 3).
>
> Regarding your point on omitting the ConfxRank notion, we would like to emphasize again that our choice of the statistic $V$ was motivated by our investigation of the miscoverage patterns in standard conformal prediction in the first place. The motivation behind our method based on trust score is that both $\textsf{Trust}$ and $\textsf{Rank}$ share the same patterns. We believe that keeping the discussion on the $\textsf{Rank}$ score will help readers better understand the rationale behind our method from first principles.
>
> > Regarding the naive approach, my point is that if your contribution were solely to define a Mondrian approach based on a new criterion—which you correctly label as naive—then your contribution would be highly incremental... I completely agree with the author regarding the limitations of the naive approach. However, the conditional approach is also not without its limitations, such as the added computational complexity.
>
> Our main contribution is that we identify a lower-dimensional statistic $V$ to capture miscoverage trends in standard conformal prediction, and improve conditional coverage with respect to $V$. We do not feel it is fair to say identifying this new criterion is incremental – we believe this is a significant contribution considering the challenges involved in achieving conditional coverage with respect to high-dimensional features. This is also noted independently by reviewers dbo9 and BLuB. The “Naive” baseline in our results corresponds to a naive approach for conditioning on $V$, and is meant to serve as an ablation to assess the contribution of the functional modeling of the relation between $V$ and the threshold $\hat{q}$ on performance.
>
> (Continued in the next comment)

---

> ### Author Response · Authors · 2024-11-30
> **Official Comment by Authors [2/2]**
>
> [Contd.] Regarding the comparison with other “Mondrian approaches”, we are only aware of Mondrian approaches that cluster the input space based on predetermined subgroups. Our approach is much more general as it can operate on any input space without the need to define a set of subgroups for group-wise conditioning. In the context of image classification, it is not clear what a sensible selection of subgroups would be. If you have suggestions regarding relevant baselines, we would be happy to include them in the updated version.
>
> “*it tends to be overly conservative in many cases while offering only minimal improvement in achieving the desired conditional coverage*” – in the specific evaluation you mention (Table 1), we would like to highlight again that Naive has access to the discrete bins we use for evaluation, hence comparing the efficiency of Naive and Conditional is not fair. Despite that, the proposed Conditional method still outperforms Naive in terms of the ConfxRank coverage gap. When we go beyond the binning-based evaluation to measure approximate $X$-conditional coverage, we see the conditional approach outperforms Naive. The utility of our proposed function class $\mathcal{F}$ is also demonstrated through results on Fitzpatrick 17k, where we outperform Groupwise (Mondrian CP) without having access to skin type group labels and with no significant increase in size.
>
> Regarding the reviewer’s comment on the computational complexity of the conditional approach, we agree that higher-dimensional function classes can increase computational cost and discuss this in the paper. **However, this is where we believe our lower two-dimensional statistic may offer an advantage over other alternative functions of the representation space** – intuitively, there is a limited range of interactions to model between the dimensions and increasing the degree beyond a threshold may result in diminishing returns (see Figure 5(b)). This is a significant consideration as the cost for computing prediction sets with the procedure in Gibbs et al. is O$(nd^2)$ for each test point, where $n$ is the number of calibration samples and $d$ is the dimension of the function class.
>
> We hope this helps clarify our motivation and the experimental choices. We are happy to continue the discussion further in case there are any other questions.

---

> ### Author Response · Authors · 2024-12-04
> **Follow-up post discussion**
>
> Thank you again for taking the time to review our paper. We hope our responses through the discussion period addressed your comments and provided necessary clarifications. As the discussion phase has concluded, we would like to request you to reconsider whether the significance of our contribution is appropriately weighted in your final assessment.
>
> Thank you,
>
> Authors

---

### Official Review · Reviewer_BLuB · 2024-11-04

**Soundness:** 3
**Presentation:** 3
**Contribution:** 2
**Rating:** 6
**Confidence:** 4

**Summary:**

The paper "Conformal Prediction Sets with Improved Conditional Coverage Using Trust Scores" presents a novel method to enhance conformal prediction by targeting conditional coverage where it is most needed. Standard conformal prediction provides a marginal guarantee on coverage but may fail to deliver meaningful coverage for individual predictions, especially for cases where a model is overconfident in incorrect predictions. To address this, the authors propose a new conformal prediction approach that incorporates a model's confidence and a nonparametric trust score as variables to approximate conditional coverage. This modified approach adapts prediction sets based on the classifier's deviation from the Bayes optimal classifier, ensuring that high-uncertainty cases receive more reliable coverage. The approach is validated through experiments on multiple datasets, including ImageNet and a clinical dataset for skin condition classification, where it improves conditional coverage properties across various subgroups.

**Strengths:**

The paper effectively addresses a significant limitation in standard conformal prediction by focusing on cases where a model's confidence may not be trustworthy. This targeted approach for conditional coverage enhances the practical utility of prediction sets, especially in high-stakes applications where overconfident errors can lead to critical consequences.

By incorporating trust scores as an indicator of the model's agreement with a Bayes-optimal classifier, the authors introduce a practical, interpretable method to approximate conditional coverage. This is a valuable innovation, providing a robust alternative to exact conditional coverage, which is often infeasible in high-dimensional settings.

The method is tested across diverse datasets, including ImageNet, Places365, and Fitzpatrick 17k, demonstrating its effectiveness in various contexts and its robustness in improving conditional coverage, even in settings with significant class imbalance or demographic subgroups.

**Weaknesses:**

The method's reliance on confidence and trust scores may limit its applicability in settings where these measures do not align well with true conditional coverage. Trust scores may not fully capture cases where model predictions deviate from true labels, particularly in datasets with noisy or ambiguous labels.

The algorithm requires calculating trust scores and confidence measures across potentially large datasets, which may limit scalability. Additionally, higher-degree polynomial functions in the function class can increase computational costs significantly, as noted by the authors.

The chosen conditional coverage thresholds and sub-intervals for confidence and trust scores could be difficult for practitioners to interpret or set in a new application context. More practical guidance on selecting these thresholds would improve usability.

Although the method is well-tested on image datasets, its performance in non-image domains (e.g., NLP or tabular data) is unexplored. This limits the generalizability of findings, especially since confidence and trust scores might behave differently in these contexts.

 While the use of multiple metrics provides detailed insights into conditional coverage, it may be cumbersome for practitioners to apply all of these metrics in practice, especially when assessing model performance quickly.

**Questions:**

How would you recommend adapting this approach if confidence or trust scores are not reliable indicators of miscoverage in a specific domain?


Could you provide more detail on how the bins for confidence and trust scores are chosen? Are there recommended strategies for selecting bin thresholds for different datasets?

Have you considered evaluating the method on datasets from non-image domains, such as text classification or structured tabular data? How do you expect the trust score approach to generalize to these contexts?

Have you performed any benchmarks on the computational complexity of the method, particularly when calculating trust scores and confidence thresholds in high-dimensional datasets?

---

> ### Author Response · Authors · 2024-11-22
> **Official Response by Authors [1/2]**
>
> Thank you for your review, and for acknowledging the effectiveness of our method in addressing a significant challenge in standard conformal prediction. We address specific questions and remarks below:
>
> > W1/Q1: Regarding applicability of the approach
>
> We do discuss this in the limitations of our approach. In our analysis over four large-scale datasets, we found the trust score to correlate well with the rank of the true class (Figure 1, Table 3) to study miscoverage, however it would certainly be interesting to explore this relationship more formally. That said, a formal guarantee is not necessary for the applicability of our method as the relationship between trust and rank scores can be empirically tested in any dataset. To determine the applicability of our method, we recommend testing the hypothesis of whether trust scores are an indication of the rank on a subset of held out data and make a decision based on the correlation and p-values.
>
> > W2/Q4: Regarding scalability and computational complexity
>
> Confidence measure is the top (max) softmax output of the classifier, which is readily available from a pretrained model. For calculating the nearest neighbor distance to each class for the trust scores computation, we use IndexFlatL2 from FAISS, Meta’s open-sourced GPU-accelerated library for efficient similarity search. This reduces the single nearest neighbor search time to ~ 0.06 ms/sample (averaged over 10 runs) on ImageNet on a single Nvidia GeForce GTX 1080. We thank the reviewer for this point and have updated the manuscript to add this in Appendix B.3.
>
> Regarding the reviewer’s second point, we agree that higher-dimensional function classes can increase computational cost and discuss this in the paper as you mentioned. **However, this is where we believe our lower two-dimensional statistic may offer an advantage over other alternative functions of the representation space** – intuitively, there is a limited range of interactions to model between the dimensions and increasing the degree beyond a threshold may result in diminishing returns (see Figure 5(b)). To compute the prediction sets for conditional conformal procedure, we follow Gibbs et al., 2023. The computational complexity of this procedure is obtained as follows: for the linear program during calibration, $O((n+d)^{1.5}n)$ is a reasonable bound for the runtime, where $n$ is the number of calibration samples and $d$ is the dimension of the function class; to update the fit for each test point, an iterative algorithm is run with a small number of iterations in practice, and hence the cost for each test point can be estimated as $O(nd^2)$, which is the cost per iteration. We also check this with the authors.
>
> > W3: Regarding interpretability of coverage thresholds and sub-intervals
>
> We would appreciate more clarification here. The thresholds are not selected manually but are computed by the conformal procedure. While the interpretability of prediction sets themselves is desirable for usability, it is not clear why we would require the thresholds to be interpretable. In an ideal scenario where we have a consistent estimate of the true conditional quantile function of $Y | X$, every sample may have a different threshold. We are happy to discuss this further based on the reviewer’s response.
>
> > W4/Q3: Regarding generalizability to non-image domains
>
> It would be an interesting direction to explore the empirical performance on non-image domains.  That said, there does not seem to be any conceptual reason why these scores will not work in other domains. The original work on trust scores [1] includes extensive evaluation on tabular datasets; additionally, there has been evidence in other works of nearest neighbor based scores working well for image as well as text domains [2, 3].
>
> > W5: Regarding applying multiple evaluation metrics
>
> We do not believe this is a limitation of our work – in fact, we choose to include multiple metrics to provide a comprehensive evaluation of conditional coverage. Regarding the reviewer’s point, once the prediction sets are computed, it is not expensive to evaluate the performance on multiple metrics. The practitioner can choose to evaluate on one or multiple metrics based on the conditional coverage notion important for their specific application.

---

> > ### Author Response · Authors · 2024-11-22
> > **Official Response by Authors [2/2]**
> >
> > > Q2: Regarding selection of confidence and trust score bins
> >
> > Certainly. As we mention in Appendix B.1, we split the samples into evenly spaced bins based on confidence and then split each confidence bin into equal-size bins based on Trust score. This is to avoid grouping samples with vastly different confidence values together. Prior work on confidence calibration has considered both bin intervals with equal number of samples (i.e., quantile-based) as well as equally-spaced bin intervals. We are not aware if there is a single recommended strategy. We choose our specific binning to have appreciable granularity while also ensuring most bins have sufficient numbers of samples in all cases. We experimented with different binning strategies as well as the number of bins, and did not find this to change the trend in our results.
> >
> > Beyond this evaluation, we also include evaluation that is independent of binning-based decisions – approximate $X$-conditional coverage evaluation (Figure 2) and evaluation on Fitzpatrick 17k (Section 4.4), where the bins are defined by skin type groups.
> >
> > [1] Jiang, H., Kim, B., & Gupta, M.R. (2018). To Trust Or Not To Trust A Classifier. Neural Information Processing Systems.
> >
> > [2] Xiong, M., Deng, A., Koh, P., Wu, J., Li, S., Xu, J., & Hooi, B. (2023). Proximity-Informed Calibration for Deep Neural Networks. Neural Information Processing Systems.
> >
> > [3] Yuksekgonul, M., Zhang, L., Zou, J.Y., & Guestrin, C. (2023). Beyond Confidence: Reliable Models Should Also Consider Atypicality. Neural Information Processing Systems.

---

> ### Author Response · Authors · 2024-11-29
> **Requesting Review of Author Response**
>
> Thank you again for taking the time to review our paper. As we approach the end of the discussion phase, we would like to confirm if our responses addressed your comments. In case there are any remaining questions or clarifications required, we are happy to discuss further.
>
>
> Thank you,
>
> Authors

---

> > ### Comment · Reviewer_BLuB · 2024-12-02
> >
> > Thanks for addressing mmost of the concerns raised in the review. I maintain my positive score of acceptance for the paper.

---

### Official Review · Reviewer_dbo9 · 2024-11-10

**Soundness:** 3
**Presentation:** 3
**Contribution:** 3
**Rating:** 6
**Confidence:** 4

**Summary:**

The paper focuses on the issue of covariate-conditional properties of conformal prediction methods. They build upon the method introduced by [Gibbs et. al. 2023] by introducing a function class that can be optimized for improving conditional validity. The main guiding principle is to estimate a low dimensional function from data such that it indicates the covariates that are more likely to have poor uncertainty quantification. They propose the discrepancy between the bayes optimal classifier and the predictor at hand as a proxy for poor uncertainty quantification, hence they build their function class based on trust scores which captures this discrepancy. They also have some empirical evidence for the good performance of their approach.

**Strengths:**

- The issue of conditional validity is central in study of conformal prediction methods. This issue particularly becomes severe in high dimensions (input dimension), which is the exact focus of the paper where they try to learn low dimensional signals as proxies for poor uncertainty quantification.
- The paper is well-explained and well-positioned within the recent advancements in the CP community.

**Weaknesses:**

- Despite the motivations, it is not clear if the notion of trust scores can effectively capture the complexities of miscoverages of different covariate points. In particular, it would have been very helpful to see some theory, at the very least in the large data regimes, that the trust scores are sufficient to capture (at least approximately) conditional coverage. The field of conformal prediction has always been developed with strong theoretical underpinnings that ensures reliable uncertainty quantification.
- With the lack of theoretical support, the extent of empirical evaluation is very important. I believe there should be a larger scale comparison with other existing approaches in the literature. For instance, one straightforward method that is an immediate rival to yours is to pick the function class of Gibbs et. al. to be a linear head on top of the representation layer of the trained classifier. Similarly, one can also use a linear head on top of any foundational model (like CLIP). These methods also bring down the dimension of the problem to low dimensional rep. layer and might actually work very well in practice.
- In regards of related works coverage, there are some holes that are good to be covered. For instance, this idea of learning features from the data (like trust scores) that are beneficial for conditional coverage is not entirely new. For instance, https://arxiv.org/abs/2404.17487 also looks at the problem from the same angle, and this is just an example! There should be at least a paragraph to cover works like this and distinguish the difference in their methodologies and yours.

**Questions:**

No further question. Might ask some after seeing the authors response.

---

> ### Author Response · Authors · 2024-11-22
> **Official Response by Authors [1/2]**
>
> Thank you for your review, and in particular, for acknowledging the significance of the focus of our paper. We are glad you found the paper well-explained and positioned within the recent progress in the conformal prediction community. We address the specific remarks below:
>
> > Regarding the ability of trust scores to capture conditional miscoverage patterns
>
> The $\textsf{Rank}$ variable, by definition, inherently captures miscoverage patterns, as miscoverage events are characterized by a high $\textsf{Rank}$ for the true label and low confidence—a relationship we empirically validate in Figure 1. For both theoretical and empirical reasons, we use the $\textsf{Trust}$ score as a proxy for the $\textsf{Rank}$ score:
>
> From a **theoretical** perspective: Theorem 4 in Jiang et al., 2018 [1] demonstrates that, under certain regularity assumptions, the $\textsf{Trust}$ score can indicate whether the model $f(X)$ aligns with the Bayes-Optimal model $f^*(X)$ as follows:
>
> $\textsf{Trust}(X) < \gamma \Rightarrow f(X) \neq f^*(X),$
>
> $1/\textsf{Trust}(X) < \gamma \Rightarrow f(X) = f^*(X),$
>
> for some threshold $\gamma$. This result suggests that the $\textsf{Trust}$ score stratifies prediction instances based on their alignment with the Bayes-Optimal classifier. In essence, it provides a coarser analog to the binning in Figure 1 into $\textsf{Rank}=1$ and $\textsf{Rank}>1$ categories. Since the $\textsf{Trust}$ score evaluates agreement with the Bayes-Optimal classifier through thresholding, it is intuitive that higher continuous values of the $\textsf{Trust}$ score correlate with increasing $\textsf{Rank}$ scores.
>
> From an **empirical** perspective: While we have not formally established a theoretical one-to-one or monotonic relationship between $\textsf{Trust}$ and $\textsf{Rank}$, Figures 1, 3, 4, and 6 provide empirical evidence supporting this intuitive relationship.
>
> We appreciate the reviewer’s insightful comment and will include a relevant discussion in Section 3 of the revised manuscript.
>
> > Regarding the expansion of our empirical evaluation
>
> We would like to highlight that our main contribution is the selection of new input dimensions that can operate in any function class to approximate conditional coverage, and not a specific choice of a function class. This was motivated by our investigation of the miscoverage patterns in standard conformal prediction in the first place – in particular, to improve coverage in instances where a classifier is *overconfident in its incorrect predictions* (as can be seen from Figures 3, 4, Appendix C.1). With the confidence and $\textsf{Trust}$ scores as input dimensions, there are possible alternative function classes other than polynomial functions that may work well in practice.
>
> It is also possible to investigate input dimensions other than confidence and $\textsf{Trust}$ as baseline. This includes the representation layers of classifiers as suggested by the reviewer. The additional advantage of our method over alternative functions of the representation space is that we propose a lower *two-dimensional statistic* to capture the miscoverage trends  – intuitively, there is a limited range of interactions to model between the dimensions and increasing the degree beyond a threshold may result in diminishing returns (see Figure 5(b)). This is a significant consideration as the cost for computing prediction sets with the procedure in Gibbs et al. is $O(nd^2)$ for each test point, where $n$ is the number of calibration samples and $d$ is the dimension of the function class. We face computational limitations as we increase $d$ beyond 21 (i.e., degree > 5), given the scale of our datasets.
>
> **That said, we appreciate your suggestion and have included further evaluation in our work (Appendix C.4)**. We implement a competitive approach using the top principal components of the feature layer as our function class. We choose the number of principal components as 20, with an added intercept term to achieve marginal coverage. Performance as evaluated by all our metrics, including approximate $X$-conditional coverage shows that our method consistently outperforms this function class. We add the results table below and include the figure with approximate $X$-conditional coverage evaluation in the paper (Appendix C.4, Figure 7). We could not implement additional function classes due to the time and computational constraints during the rebuttal, but are happy to include in the final version based on further suggestions.

---

> ### Author Response · Authors · 2024-11-22
> **Official Response by Authors [2/2]**
>
> | Dataset | Method | Marginal |  | Conditional |  |  |
> |---|---|---|---|---|---|---|
> |  |  | Coverage | Size | Conf x Trust CovGap | Conf x Rank CovGap | Class-conditional CovGap |
> | ImageNet | STANDARD | 0.90 (0.00) | 4.32 (0.04) | 6.35 (0.11) | 33.12 (0.15) | 7.23 (0.04) |
> |  | NAIVE | 0.93 (0.00) | 7.21 (0.05) | 8.01 (0.50) | 23.82 (0.15) | 6.68 (0.02) |
> |  | CONDITIONAL | 0.90 (0.00) | 22.36 (0.83) | **4.37** (0.09) | **23.66** (0.24) | **6.02** (0.04) |
> |  | CONDITIONAL (PCA) |  0.90 (0.00) | 6.54 (0.07) | 6.23 (0.09) | 31.27 (0.14) | 6.92 (0.04)|
> | ImageNet-LT | STANDARD     | 0.89 (0.00) | 50.35 (0.02)| 4.47 (0.02) | 19.92 (0.03) | 8.32 (0.00) |
> |                 | NAIVE        | 0.89 (0.00) | 46.63 (0.07)| 2.99 (0.01) | 18.27 (0.02) | 8.29 (0.01) |
> |                 | CONDITIONAL  | 0.90 (0.00) | 58.64 (0.18)| **2.21** (0.01) | **17.09** (0.02) | **7.92** (0.00) |
> |  | CONDITIONAL (PCA) |  0.89 (0.00) | 53.59 (0.05) | 4.78 (0.01) | 20.13 (0.04) | 8.17 (0.01) |
> | Places365   | STANDARD     | 0.90 (0.00) | 14.17 (0.07)| 5.78 (0.08) | 25.58 (0.09) | **4.99** (0.05) |
> |                 | NAIVE        | 0.90 (0.00) | 12.76 (0.11)| **2.40** (0.16) | 22.22 (0.13) | 5.11 (0.06) |
> |                 | CONDITIONAL  | 0.90 (0.00) | 15.98 (0.57)| 4.38 (0.10) | **22.09** (0.12) | **4.98** (0.05) |
> |  | CONDITIONAL (PCA) | 0.90 (0.00) | 13.89 (0.07) | 5.34 (0.07) | 25.17 (0.12) | **4.96** (0.04) |
> | Places365-LT| STANDARD     | 0.90 (0.00) | 43.46 (0.10)| 5.55 (0.02) | 13.23 (0.01) | **5.34** (0.00) |
> |                 | NAIVE        | 0.90 (0.00) | 38.54 (0.06)| 4.55 (0.02) | **11.75** (0.01) | 5.61 (0.01) |
> |                 | CONDITIONAL  | 0.90 (0.00) | 37.07 (0.03)| **1.72 (0.00)** | **11.75** (0.01) | 5.65 (0.00) |
> |  | CONDITIONAL (PCA) | 0.90 (0.00) | 45.03 (0.13) | 5.37 (0.03) | 12.89 (0.01) | **5.35** (0.01) |
>
>
> > Regarding the inclusion of more related work
>
> Thank you for pointing us to Kiyani et al., 2024. We have added this discussion to our related work, including other works that share similar motivation.
>
> [1] Jiang, H., Kim, B., & Gupta, M.R. (2018). To Trust Or Not To Trust A Classifier. Neural Information Processing Systems.

---

> > ### Comment · Reviewer_dbo9 · 2024-11-25
> >
> > I have read the authors response and the other reviewers comments. Most of my concerns are addressed and I believe the paper contributions are enough for acceptance. In particular, I find the idea of designing\learning low dimensional signals as proxies for conditional miscoverage a very promising research direction. Hence, I increase my score and vote for acceptance.

---

### Meta-Review · Area_Chair_EFCR · 2024-12-24

**Metareview:**

The paper proposes a conformal prediction method targeting conditional coverage by incorporating model confidence and trust scores as proxies for rank-based conditional patterns. While reviewers acknowledged its clarity and relevance in addressing challenges of overconfident mispredictions, significant concerns were raised about its theoretical foundations, experimental evaluations, and overall practical impact. Multiple reviewers noted the absence of strong theoretical support for the effectiveness of trust scores in capturing conditional miscoverage patterns.

**Additional Comments On Reviewer Discussion:**

While the paper addresses an important problem and provides a practical approach to conditional coverage, the concerns over theoretical rigor, incremental contributions, limited empirical scope, and trade-offs between prediction set size and coverage outweigh its strengths. The reviewers appreciated the clarity and motivation but did not lean toward a clear accept.

---

### Decision · Program_Chairs · 2025-01-22

Reject